

# Studying the impact of biomass burning aerosol radiative and climate effects on the Amazon rainforest productivity with an Earth System Model

Florent F. Malavelle[1], Jim M. Haywood[1,2], Lina M. Mercado[3,4], Gerd A. Folberth[2], Nicolas Bellouin[5],
Stephen Sitch[3] and Paulo Artaxo[6]

[1]CEMPS, University of Exeter, Exeter, EX4 4QE, UK
[2]UK Met-Office Hadley Centre, Exeter, EX1 3PB, UK
[3]CLES, University of Exeter, Exeter, EX4 4RJ, UK
[4]Centre for Ecology and Hydrology, , OX10 8BB, UK
[5]Department of Meteorology, University of Reading, Reading, RG6 6BB, UK
[6]Department of Applied Physics, Institute of Physics, University of Sao Paulo, Sao Paulo, Brazil

*Correspondence to*: Florent F. Malavelle (f.malavelle@exeter.ac.uk)

**Abstract.** Diffuse light conditions can increase the efficiency of photosynthesis and carbon uptake by vegetation canopies.
The diffuse fraction of photosynthetically active radiation (PAR) can be affected by either a change in the atmospheric
aerosol burden and/or a change in cloudiness. During the dry season, a hotspot of Biomass Burning on the edges of the
Amazon rainforest emits a complex mixture of aerosols and their precursors and climate-active trace gases (e.g. $CO_2$, $CH_4$,
$NO_x$ etc). This creates potential for significant interactions between chemistry, aerosol, cloud, radiation and the biosphere
across the Amazon region. The combined effects of biomass burning on the terrestrial carbon cycle for the present-day are
potentially large, yet poorly quantified. Here, we quantify such effects using the Met Office Hadley Centre Earth System
Model HadGEM2-ES which provides a fully coupled framework with interactive aerosol, radiative transfer, dynamic
vegetation, atmospheric chemistry and biogenic volatile organic compound emission components. Results show that the
overall net impact of present-day biomass burning aerosols is to increase net primary productivity (NPP) by +80 to +105
TgC/yr, or 1.9 to 2.7%, over the central Amazon basin on annual mean. For the first time we show that this enhancement is
the net result of multiple competing effects: an increase in diffuse light which stimulates photosynthetic activity in the
shaded part of the canopy (+65 to +110 TgC/yr), a reduction in the total amount of radiation (-52 to -105 TgC/yr) which
reduces photosynthesis and feedback from climate adjustments in response to the aerosol forcing which increases the
efficiency of biochemical processes (+67 to +100 TgC/yr). These results illustrate that despite a modest direct aerosol effect
(the sum of the first two counteracting mechanisms) the overall, net impact of biomass burning aerosols on vegetation, is
sizeable, when indirect climate feedbacks are considered. We demonstrate that capturing the net impact of aerosols on
vegetation should be assessed considering the system-wide behaviour.





## 1 Introduction

The Amazon rainforest is the largest expanse of tropical forest on Earth. It provides invaluable ecological services and plays a major role in the Earth system and climate (Malhi *et al.*, 2000). The Amazon rainforest is a net sink of atmospheric $CO_2$ although drought frequency and intensity which are expected to increase in the future could have severe consequences for

future forest resilience, and potentially shift the Amazon rainforest from a sink to a net source of atmospheric $CO_2$ (Cox *et al.*, 2000, 2004; Phillips *et al.*, 2009; Duffy *et al.*, 2015; Doughty *et al.*, 2015; Sakschewski *et al.*, 2016; Zemp *et al.*, 2017). This possibility motivated intense research to develop a better understanding of the rainforest response to environmental stresses via integrated explicit representations of the carbon cycle in Earth System Models (ESM) (Cox *et al.*, 2000). Response to many of these environmental stresses is now well documented and represented in ESM's, including the effects

of surface temperature, atmospheric composition, water availability or the amount and quality of accessible light (direct versus diffuse) for plant photosynthesis (e.g. Nemani *et al.*, 2003; Cox *et al.*, 2008; Sitch *et al.*, 2007; Mercado *et al.*, 2009; Beer *et al.*, 2010; Ciais *et al.*, 2013; Pacifico *et al.*, 2015; Unger *et al.*, 2017).

In parallel to the above-mentioned environmental stresses, forest fires are also an intrinsic component of some forest

lifecycles, providing an additional mechanism for depleting land carbon reservoirs. Intense biomass burning events present a notorious pressure on tropical regions which typically occur during the dry season – i.e. between around August and September in the Amazon region (Artaxo *et al.*, 2013, Brito *et al.*, 2014). Fires in general occur naturally, however, a significant fraction results from the anthropogenic pressure that continually erode the fragmented forest edges (Cochrane, 2003). Despite a decreasing trend in the rate of deforestation over the last decade as a result of stricter environmental policies

(Kalamandeen *et al.*, 2018), it is estimated that 293 Teragrams of Carbon per year (TgC/yr) are directly released back into the atmosphere from fires in the Amazon (van der Werf *et al.*, 2006). Fires can also have an indirect impact on the rainforest carbon budget that is harder to quantify; for instance, fires alter surface properties (e.g. albedo) in the burnt area which can modify surface fluxes and the water cycle (e.g. Spracklen *et al.*, 2012). Additionally, fires emit a complex mixture of gases ($CO_2$, $CO$, $CH_4$, $NO_x$ and VOCs), aerosols and aerosol-precursors which can affect remote regions of the rainforest after

being dispersed by the wind. Pacifico *et al.* (2015) illustrated such a mechanism by analysing the potentially harmful effect of near-surface ozone ($O_3$) associated with biomass burning and estimated that the rainforest gross primary productivity (GPP) was reduced by up to approximately -230 TgC/yr, a number of similar magnitude to the magnitude of the direct carbon loss from fires.

Assessing the overall impact of Amazonian forest fires on ecosystems is challenging as it encompasses a combination of direct losses, and indirect impacts from the fire by-products which can depend on intricate interactions between the biosphere, atmospheric composition, radiation and energy budget, clouds and the water cycle to cite a few (Bonan 2008).



Here, we aim to specifically elucidate the impact of biomass burning aerosols (BBA) that are associated with forest fires and quantify their potential effect on the Amazon forest productivity.

Significant amounts of BBA are emitted in South America which strongly modify the radiative budget by scattering and absorbing solar radiation. This reduces the level of Photosynthetically Active Radiation (PAR), traditionally defined as the radiation between wavelengths of 300 and 700 nm, reaching the surface and used by plants to photosynthesize (i.e. to assimilate carbon from the atmosphere). Contrary to intuition, an increase in the diffuse light fraction can be beneficial to plants as the shaded, non-light-saturated leaves, typically found in the understory or lower canopy layers, receive more radiation under diffuse light conditions than they would normally experience under direct light conditions owing to shading by leaves fully exposed to sunlight. As a result, this trade-off between experiencing less PAR overall but receiving more evenly distributed light across the canopy favours higher rates of canopy photosynthesis. The first comprehensive estimation of this Diffuse PAR Fertilization Effect (DFE) at the global scale was documented by Mercado *et al.* (2009a), who used a combination of offline aerosol distributions, radiative transfer and a land surface model to estimate that DFE may have increased the global land carbon uptake by up to 25% during the global dimming period (1950-1980; Stanhill and Cohen, 2001). More recently, Rap *et al.* (2015) used a similar framework of offline models to assess the role of BBA over the Amazon region. They showed that BBA increase the annual mean diffuse light and net primary productivity (NPP) by 3.4–6.8% and 1.4–2.8%, respectively. Strada and Unger (2016) took a step further using a coupled modelling framework to estimate biomass burning aerosol impacts on Amazon forest GPP, obtaining an increase of 2–5% on annual means. Recently, Moreira *et al.* (2017) also applied a coupled framework using a regional model (BRAMS) to conclude that BBA could increase the GPP of the Amazon forest by up to 27% during the peak of the biomass burning season. The study of Moreira *et al.* (2017) assumed high BBA emissions and did not accounted for the effect of cloudiness on the diffuse fraction of radiation, so it provides an upper estimate of the potential impact of the effects of the attenuation of total solar radiation and the enhancement of the diffuse solar radiation flux inside the vegetation canopy.

Despite a growing body of evidence supporting the DFE mechanism, both from observational and modelling perspectives (e.g. Cohan *et al.*, 2002; Gu *et al.*, 2003; Robock *et al.*, 2005; Yamasoe *et al.*, 2006; Mercado *et al.*, 2009a; Kanniah *et al.*, 2012; Cirino *et al.*, 2014; Cheng *et al.*, 2015), a full quantification of BBA impact on ecosystems remains poor because aerosols-radiation interaction (ARI), and to some extent aerosol-cloud interactions (ACI), not only create the conditions for a DFE but also modify the climate locally. For example, a regional haze of aerosols can perturb regional hydroclimates (Nigam and Bollasina 2010), force clouds to adjust to aerosol 'semi-direct' and 'indirect' effects which modify the way clouds interact with radiation (Hansen *et al.*, 1997; Haywood and Boucher, 2000; Koren *et al.*, 2004), or create a positive cooling effect on productivity by reducing surface heat stress in hot environments, allowing for more efficient uptake of atmospheric $CO_2$ through leaf stomata (Robock *et al.*, 2005; Xue *et al.*, 2016; Strada and Unger, 2016). Neglecting such essential coupling pathways may overemphasise the relative contribution of the DFE due to loss of internal consistency that



do not allow variability within non-linear relationships. To our knowledge, only two studies (Strada and Unger, 2016; Unger *et al.*, 2017) have considered the DFE within a fully coupled earth system framework (using the NASA GISS ModelE2–YIBs) to investigate the role of aerosols and haze on vegetation. Although these studies have investigated the role of diffuse radiation on GPP and isoprene emissions (Strada and Unger, 2016; Unger *et al.*, 2017), none of the existing studies has

separately quantified the contribution from climate and radiative effects from aerosols in order to make an assessment of the net effects from aerosols on vegetation productivity. In the present study, we apply an ESM modelling framework to quantify the impact of present-day BBA via quantification of individual and net effects of changes in diffuse radiation, direct radiation and climate upon the vegetation productivity in the Amazon rainforest. For this endeavour, we have implemented an updated representation of plant photosynthesis and carbon uptake that is sensitive to diffuse light radiation in the UK Met

Office HadGEM2-ES Earth System Model (Mercado *et al.* 2007, 2009a). In addition, a framework that disentangles the vegetation response has been developed to provide a deeper understanding of the contributions of different plant environmental variables affected by aerosols. The role of $O_3$ precursor emissions and in-situ formation of $O_3$ associated with biomass burning (Pacifico *et al.*, 2015) are not considered here.

The methodology and the experimental setup are described in Sect. 2. Results are discussed in Sect. 3, including first a model evaluation in Sect. 3.1, then the net effect of BBA in Sect. 3.2 and individual contributions from the diffuse light fraction, the reduction in total PAR and the *climate feedbacks* associated with the BBA perturbation in Sect. 3.3. These findings are contextualized in Sect. 3.4 by analysing the results from four additional sensitivity experiments designed to elucidate the role of aerosol optical properties, aerosol-cloud interactions, the atmospheric $CO_2$ concentration and vertical

distribution of nitrogen through the canopy. Concluding remarks and a summary of this study's main results are provided in Sect. 4 and Sect. 5, respectively.

## 2 Method

We evaluate the effects of biomass burning aerosol-radiation interactions upon the Amazon rainforest primary productivity for present-day conditions using the Met Office Hadley Centre Global Environment Model HadGEM2-ES (The HadGEM2

Development Team, 2011) which provides a fully coupled framework. The model is briefly described in Sect. 2.1.

We present the results of a sensitivity experiment (Sect. 3) which consists of varying the biomass burning aerosol emissions only over South America. 'Real world' fires also emit greenhouse gases (e.g. $CO_2$, CO, $CH_4$) and ozone precursors ($NO_x$, VOC) which can potentially affect the biosphere. Ozone is particularly critical as it is a pollutant which harms plants and

reduces their productivity, thus their ability to draw $CO_2$ from the atmosphere (Sitch *et al.*, 2007). Whereas the damaging effect of ozone is not accounted for in this study, we will briefly discuss the potential fertilization effect from increased $CO_2$



background that can result from biomass burning in Sect. 4. The ozone damage effect has been documented by Pacifico *et al.* (2015) using a similar modelling framework as in the present study and we refer readers to that study for further details.

Atmospheric particles such as aerosols and cloud droplets scatter radiation which increases the fraction of radiation that is

diffuse. Diffuse conditions result in higher light use efficiency of plant canopies which can enhance carbon uptake (Roderick *et al.*, 2001; Gu *et al.*, 2002). An increase in diffuse radiation is concomitant with a decrease in the overall amount of radiation (Supplementary Fig. S1). These two opposing effects will be referred to in the rest of the manuscript as *change in diffuse fraction* and *reduction in total PAR,* respectively and will be quantified separately in Sect. 3.3. Finally, BBA effects impact the coupled system which itself controls the rate of biochemical processes of vegetated land surfaces. We will simply

refer to these adjustments to the BBA effects as *climate feedback* in the remainder of the manuscript. The sum of '*climate feedback'*, '*change in diffuse fraction'* and '*reduction in total PAR'* is referred as the *net impact* of BBA on plant productivity. The framework we developed to disentangle these three terms is described in Sect. 2.4.

## 2.1 Model description

HadGEM2-ES is an earth system model built around the HadGEM2 atmosphere-ocean general circulation model and

includes a number of earth system components such as:

- the ocean biosphere model diat-HadOCC (Diatom-Hadley Centre Ocean Carbon Cycle), developed from the HadOCC model of Palmer and Totterdell (2001).
- the sea-ice component (The HadGEM2 Development Team, 2011).
- the Top-down Representation of Interactive Foliage and Flora Including Dynamics (TRIFFID) dynamic global

vegetation model (Cox, 2001), and the land-surface and carbon cycle model MOSES2, collectively known as JULES (Met Office Surface Exchange Scheme; Cox *et al.*, 1998, 1999; Essery *et al.*, 2003).

- the interactive Biogenic Volatile Organic Compounds (iBVOC) emission model (Pacifico *et al.*, 2012).
- the UKCA tropospheric chemistry (O'Connor *et al.*, 2014).

The atmospheric model resolution is N96 (1.875° by 1.25°) with 38 vertical levels with the model top at ~39 km. Our

modelling framework is similar to the configuration used by Pacifico *et al.* (2015) who provide a detailed analysis of the successful model performance against observations.

For clarity, we provide some additional details on the treatment of aerosols and their coupling with radiation and clouds and on the updated representation of the canopy interaction with radiation. The radiative transfer code in the atmospheric part of HadGEM2-ES is SOCRATES (Edwards and Slingo, 1996), which parametrises radiative fluxes using a 'two-stream'

approximation (Meador and Weaver, 1980). The radiative transfer is solved for 6 wavebands in the shortwave and 9 in the



longwave. This scheme accounts for radiation interaction with aerosol particles by defining 3 single scattering properties on a layer: optical depth, single scattering albedo (the ratio of scattering efficiency to total extinction) and an asymmetry parameter. Together, these properties determine the overall transmission and reflection coefficients of each atmospheric layer. At the interface between the lowest atmospheric level and the land surface, the total and the direct radiances for the

short-wave band 320-690 nm, which approximates the PAR, calculated by the SOCRATES radiation scheme are transferred to the land surface routines to calculate plant photosynthesis.

In the JULES land surface model, the total and direct irradiance components of PAR calculated by the atmospheric model provide the boundary conditions at the top of the canopy. The diffuse PAR fraction is calculated as the difference between total and direct radiation, divided by the total radiation. The canopy is discretized into 10 vertical layers and the radiative

transfer in the canopy is also parametrised with a 'two-stream' approximation but using more detailed assumptions to represent light interception by foliage (Sellers, 1985). The photosynthesis model is based upon the observed processes at the leaf scale, which are then scaled up to represent the canopy. It takes into account variations in direct and diffuse radiation on sunlit and shaded canopy photosynthesis at each canopy layer. In this way, photosynthesis of sunlit and shaded leaves is calculated separately under the assumption that shaded leaves receive only diffuse light and sunlit leaves receive both diffuse

and direct radiation (Dai *et al.*, 2004; Clark *et al.*, 2011). Leaf-level photosynthesis is calculated using the biochemistry of C3 and C4 photosynthesis from Collatz *et al.* (1991) and Collatz *et al.* (1992).

This canopy radiation scheme was first developed to quantify the impact of anthropogenic aerosol emissions on the global carbon cycle (Mercado *et al.*, 2007, 2009a) and consequently implemented in JULES (Clark *et al.*, 2011). It is a novel addition to HadGEM2-ES as it was not available during the HadGEM2-ES contribution to CMIP5. HadGEM2-ES with the

previous canopy radiation scheme had a tendency to overestimate GPP (Shao *et al.*, 2013), which has to be balanced by high plant respiration (RESP) to get satisfactory estimates of global NPP (i.e. NPP=GPP-RESP). The new representation of light interception that we have implemented is able to reproduce higher light use efficiency (LUE) under diffuse light conditions (Sect. 3.1 and Supplementary Fig. S2). However, the ratio of GPP to plant respiration in HadGEM2-ES with the new canopy radiation model remains too high when compared to observationally-based estimates (e.g. Luyssaert *et al.*, 2007). To correct

this deficiency, we decreased the ratio of Nitrogen allocated in the roots relative to the Nitrogen in the leaves from 100% to 50% (Clark *et al.*, 2011, Table 2 therein). Additionally, we reduced the leaf dark respiration coefficient that relates leaf dark respiration and $V_{cmax}$ from 15% to 10% (Clark *et al.*, 2011, eq. 13 therein). These changes are based on a sensitivity analysis that we performed with the stand-alone version of JULES. We used the meteorological observations from the tropical French Guyana site (assumed to be fully covered by broadleaf trees) to drive JULES and investigate the sensitivity to parameters

such as the leaf nitrogen content at canopy top ($N_{L0}$), the dark respiration coefficient and the nitrogen allocation throughout the canopy via the value of the nitrogen profile extinction coefficient (Clark *et al.*, 2011, eq. 33 therein and Sect. 2.3.4 of the present study). Fast carbon fluxes (GPP, RESP and NPP) were calculated at a 3hourly temporal resolution by varying one of



these 3 parameters individually (Supplementary Fig. S3a,b,c) and then averaged to annual mean values (Supplementary Fig. S3d,e,f). The annual means were then used to construct contour surfaces for the fast carbon fluxes by varying combinations of the selected parameters (Supplementary Fig. S4). This method enables us to ultimately pre-calibrate the fast carbon fluxes in the HadGEM2-ES model offline.

Aerosols are represented by the CLASSIC aerosol scheme (Bellouin *et al.*, 2011) which is a one-moment mass prognostic scheme. This aerosol module contains numerical representation of up to eight tropospheric aerosol species. Here, ammonium sulphate, mineral dust, sea salt, fossil fuel black carbon (FFBC), fossil fuel organic carbon (FFOC), biomass burning aerosols and secondary organic (also called biogenic) are considered. Dust and sea-salt are from diagnostic schemes based

on the near-surface windspeed, while other emissions including biogenic aerosols are represented by a relatively simple climatology (Bellouin et al., 2011). Transported species experience boundary layer and convective mixing are removed by dry and wet deposition. Wet deposition by large-scale precipitation is corrected for re-evaporation of precipitation: tracer mass is transferred from a dissolved mode to an accumulation mode in proportion to re-evaporated precipitation. For convective precipitation, accumulation mode aerosols are removed in proportion to the simulated convective mass flux.

Emissions of biomass burning aerosols are the sum of the biomass burning emissions of black and organic carbon. Grass fire emissions are assumed to be located at the surface, while forest fire emissions are injected homogeneously across the boundary layer (0.8–2.9 km).

The direct radiative effect due to scattering and absorption of radiation by all eight-aerosol species represented in the model is included. The semi-direct effect, whereby aerosol absorption tends to change cloud formation by warming the aerosol

layer, is thereby included implicitly. Wavelength-dependent specific scattering and absorption coefficients are obtained using Mie calculations from prescribed size distributions and refractive indices. All aerosol species except mineral dust and fossil fuel black carbon are considered to be hydrophilic, act as cloud condensation nuclei, and contribute to both the first and second indirect effects on clouds, treating the aerosols as an external mixture. Jones *et al.* (2001) detail the parameterization of the indirect effects used in HadGEM2-ES. The cloud droplet number concentration (CDNC) is

calculated from the number concentration of the accumulation and dissolved modes of hygroscopic aerosols. For the first indirect effect, the radiation scheme uses the CDNC to obtain the cloud droplet effective radius. For the second indirect effects, the large-scale precipitation scheme uses the CDNC to compute the auto-conversion rate of cloud water to rainwater (Jones *et al.*, 2001).

## 2.2 Experimental design: main experiment

The HadGEM2-ES model is initiated on the 1st of Dec 2000 from a previous historical simulation. We consider the year 2000 to be a good surrogate for present-day climate which will enable to assess the impact of present-day BBA emissions on vegetation. As historical simulations are transient climate simulations we constrain the carbon cycle to present-day values as




well (to be described in the next paragraph). The model is then integrated for a period of 40 years using periodic forcing for the year 2000 to construct an ensemble that captures the model internal variability. Results reported here are the multi-annual means over the final 30-years of the model integration. The domain of analysis is defined by the coordinates EQ-15°S / 70°W-53°W and is primarily covered by broadleaf tree for this configuration of HadGEM2-ES (Supplementary Fig. S5).

The HadGEM2-ES model is set-up in an Atmospheric Model Intercomparison Project (AMIP, Jones *et al.*, 2011) type configuration using prescribed climatologies of monthly mean Sea Surface Temperatures (SST) and Sea Ice Cover (SIC) which enables to analyse the rapid adjustments of land-surface climate to aerosol radiation perturbation. The introduction of a new canopy radiation interaction model introduces a significant departure in the carbon cycle balance. To prevent the need

of a complex spin-up exercise, we prescribe the vegetation cover and carbon reservoirs to present-day level. This is achieved by reducing the call frequency of the TRIFFID dynamic vegetation model to 30 years in order to maintain the vegetation in a steady state. A similar approach is discussed in Strada and Unger (2016). Overall, this enables us to focus our analysis on the fast carbon flux responses (i.e. NPP, GPP) and their sensitivity to the perturbation induced by the biomass burning aerosols.

Aerosol and their precursor emissions are taken from the CMIP5 inventories (Lamarque *et al.*, 2010). We use the decadal mean emissions centred around the year 2000 representative of present-day emissions. Biogenic Volatile Organic Compound (BVOC) emissions from vegetation (Pacifico *et al.*, 2012) are sensitive to changes in plant productivity, hence sensitive to DFE. These emissions are calculated online but are not taken into account in the CLASSIC aerosols scheme. Instead, the climatology of BVOC (also called secondary organics) from CMIP5 are used. The biomass burning emissions

representative of present-day conditions are based on the GFEDv2 inventories (Van der Werf *et al.*, 2006, Lamarque *et al.*, 2010). These are the standard emission scenario for the simulation labelled as BBAx1 for the Main Experiment. A total of five simulations are conducted in the Main Experiment where the standard biomass burning aerosols emissions are varied by -100%, -50%, 0%, +100% and +300%, respectively (simulation BBAx0, BBAx0.5, BBAx1, BBAx2 and BBAx4, respectively). A multiplication factor is applied to the emission only for the BB sources over South America

(85W,40S;30W,15N). We define the control simulation as the simulation without BBA being emitted over south America (i.e. BBAx0). The changes in fast carbon fluxes are calculated as the departure from this reference simulation (e.g. $\Delta NPP_{netimpact}^{BBAx1} = NPP^{BBAx1} - NPP^{BBAx0}$ and represents the net change in NPP due to standard emissions of BBA).

**2.3 Sensitivity experiments**

In parallel to the 5 simulations for the main experiment, we have conducted the following 4 additional sensitivity

experiments to further appreciate the role of *i)* aerosol optical properties, *ii)* aerosol-cloud interactions, *iii)* the canopy nitrogen profile and *iv)* atmospheric carbon dioxide concentration. A listing of the simulations done for the main experiment and the sensitivity experiments is provided in Table 1.



| | | Main Experiment | Sensitivity Experiment: Aerosol Optical Properties | | Sensitivity Experiment: Aerosol-Cloud Interactions | | Sensitivity Experiment: Canopy Nitrogen profile | Sensitivity Experiment: Atmospheric CO₂ concentration | |
|---|---|---|---|---|---|---|---|---|---|
| | | *Impact of BBA emissions on vegetation* | *More scattering BBA* | *More absorbing BBA* | Aerosols only affect cloud effective radius (1st AIE) | No AIE | Steeper nitrogen profile in the canopy | *+25 ppm of atmospheric $CO_2$* | *+50 ppm of atmospheric $CO_2$* |
| Present-day BBA emissions over South America | **Control runs** i.e. no BBA emission | **BBAx0** | **BBAx0$_{DIFF\_OP}$** | **BBAx0$_{ABS\_OP}$** | **BBAx0$_{1stAIE}$** | **BBAx0$_{noAIE}$** | **BBAx0$_{Steep\_N}$** | **BBAx0$_{+25ppm}$** | **BBAx0$_{+50ppm}$** |
| | Half the BBA emissions | **BBAx0.5** | / | / | / | / | / | / | / |
| | Standard BBA emissions | **BBAx1** | **BBAx1$_{DIFF\_OP}$** | **BBAx1$_{ABS\_OP}$** | **BBAx1$_{1stAIE}$** | **BBAx1$_{noAIE}$** | **BBAx1$_{Steep\_N}$** | **BBAx1$_{+25ppm}$** | **BBAx1$_{+50ppm}$** |
| | Double the BBA emissions | **BBAx2** | **BBAx2$_{DIFF\_OP}$** | **BBAx2$_{ABS\_OP}$** | **BBAx2$_{1stAIE}$** | **BBAx2$_{noAIE}$** | **BBAx2$_{Steep\_N}$** | **BBAx2$_{+25ppm}$** | **BBAx2$_{+50ppm}$** |
| | Quadruple the BBA emissions | **BBAx4** | / | / | / | / | / | / | / |

**Table 1 – List of model simulations done for the five experiments.**

### 2.3.1 Aerosol Optical properties

The representation of BBA in HadGEM2-ES is based on the measurements collected during the SAFARI2000 campaign
near South Africa (Abel *et al.*, 2003; Bellouin *et al.*, 2011). It describes the size distribution of BBA as an external mixture
of two mono-modal smoke species. For the fresh smoke, a log-normal distribution with a median geometrical radius (r), r =
0.1 μm and a geometric standard deviation (σ), σ = 1.30 are assumed. For aged smoke, r = 0.12 μm and σ = 1.30. Fresh
biomass smoke is converted to aged smoke at an exponential rate assuming an e-folding time of 6 hours which typically
accounts for the ageing of the smoke plume due to condensation of chemical species (e.g. sulphate or organic compounds,
Abel *et al.*, 2003). Optical properties for the two modes are calculated a priori (i.e. offline) using Mie theory for various
levels of relative humidity (RH) to account for hygroscopic growth. These optical properties – specific extinction, absorption
coefficients and asymmetry parameter – are then prescribed in the HadGEM2-ES radiative transfer look up table of optical
properties.

BBA optical properties may vary significantly depending on the type of vegetation burnt, combustion regime and the
meteorological conditions (Reid *et al.*, 2005). Many observational campaigns since SAFARI2000 have reported more
absorbing BBA in other regions of the world (e.g. Johnson *et al.*, 2008, 2016). Even at the regional scale, variation in BBA
optical properties may occur. For example, aircraft observations in Brazil during SAMBBA show that flaming combustion





associated with Cerrado burning in the eastern regions produce more BC and less organic aerosol, and therefore a more absorbing BBA, while smouldering forest burning in the west produces less absorbing BBA (Johnson *et al*., 2016). The degree of aerosol absorption is characterised by the single Scattering Albedo (SSA) which is the ratio of aerosol scattering over aerosol extinction. BBA with low SSA (e.g. ~0.80) absorb more solar radiation than BBA with higher SSA (e.g. ~0.90).

This can have implications from the vegetation perspective as a layer made of absorbing BBA would transmit less radiation to the surface than a layer made of more scattering BBA, limiting the amount of energy available for photosynthesis. In this experiment, we investigate this aspect by varying BBA SSA by +/- 10% by scaling the specific scattering ($K_{sca}$ in m²/kg) and absorption ($K_{abs}$ in m²/kg) coefficients ($K_{sca}$ in m²/kg) directly in the look-up tables, ensuring that specific extinction remain constant. The asymmetry parameter is assumed to be unaffected. Dry BBA optical properties at 550 nm for the aged smoke

are reported in *Table* 2.

For this sensitivity experiment, the BBAx0, BBAx1 and BBAx2 simulations are re-run twice, once assuming a more absorbing BBA and once assuming a more scattering BBA (simulations labelled BBAx0$_{DIFF\_OP}$, BBAx1$_{DIFF\_OP}$ and BBAx2$_{DIFF\_OP}$ for the diffuse case and BBAx0$_{ABS\_OP}$, BBAx1$_{ABS\_OP}$ and BBAx2$_{ABS\_OP}$ for the absorbing case, respectively).

Fig. S6 in supplementary material shows how HadGEM2-ES simulates the ambient SSA of BBA (Sup. Fig 6a) and of all aerosols (Sup. Fig 6b) after modifying the BBA optical properties. Sup. Fig. 6c shows that the amount of direct PAR is unaffected as expected because of the constraint imposed on $K_{ext}$. In the higher SSA case (i.e. more diffusing BBA), the amount of Diffuse PAR reaching the surface is increased, resulting in higher amount of Total PAR which contrasts with the lower SSA case.

| | $K_{ext}$ [m²/kg] | $K_{abs}$ [m²/kg] | $K_{sca}$ [m²/kg] | SSA |
|---|---|---|---|---|
| Scattering BBA | 5.073*1e3 | 9.191*1e2 | 4.154*1e3 | 0.99 |
| Standard BBA | 5.073*1e3 | 4.575*1e2 | 4.615*1e3 | 0.91 |
| Absorbing BBA | 5.073*1e3 | 5.074*1e-1 | 5.072*1e3 | 0.82 |

**Table 2: Dry (Relative Humidity=0%) optical properties at 550 nm for the aged Smoke Biomass Burning aerosols.**

### 2.3.2 Aerosol-cloud Interactions

Clouds critically affect the amount of radiation reaching the surface (e.g. Roderick *et al.*, 2001; Cohan *et al.*, 2002; Pedruzo-Bagazgoitia *et al.*, 2017). Aerosols have the potential to alter cloud properties (i.e. how they interact with radiation, Haywood and Boucher, 2000) and hence alter surface radiation. This experiment aims to address whether aerosols can affect

vegetation productivity indirectly by interacting with clouds. Although Aerosol-Cloud Interactions (ACI) remain very challenging to represent in ESMs (Ghan *et al.*, 2016; Malavelle *et al.*, 2017), we will investigate whether the representation of these processes in the ESM used here can have a detectable impact over the region considered in this study. The BBAx0, BBAx1 and BBAx2 simulations are done twice. In the first set of simulations (labelled BBAx0_1$^{st}$AIE, BBAx1_1$^{st}$AIE and BBAx2_1$^{st}$AIE), aerosols impact on precipitation efficiency is switched off (i.e. no second aerosol indirect effect, 2$^{nd}$AIE,

through alteration of liquid water path via auto-conversion) but can still modify cloud albedo by altering the cloud droplet



effective radius (i.e. the first aerosol indirect effect, 1stAIE). In the second set of simulations (labelled BBAx0_noAIE, BBAx1_noAIE and BBAx2_noAIE), all aerosol indirect effects are switched off. As turning off AIE reverts back CDNC to prescribed values, the BBA effect on vegetation will be calculated as a difference between simulations with the same indirect effect configuration (e.g. BBAx1_1stAIE – BBAx0_1stAIE).

**2.3.3 Canopy nitrogen profile**

Photosynthesis not only requires light, $CO_2$ and water but also nutrients that are essential in the chemistry cycles of photosynthesis. Nitrogen can be considered the most critical of those nutrients and could act as a bottleneck for plant photosynthesis (e.g. Bonan, *et al.*, 2011; Ciais *et al.*, 2014; Fernández-Martínez *et al.*, 2014; Wieder *et al.*, 2015; Zaehle *et al.*, 2015; Houlton *et al.*, 2015). Optimization arguments suggest that, in order to maximise the rates of carboxylation and the

rate of transport of photosynthetic products, nitrogen resources should be allocated at the top of the canopy (i.e. a steep decrease in the nitrogen profile) where light absorption is maximum (Alton, 2007). However, observations support a more even allocation of the nitrogen resources (i.e. a shallow decrease in the nitrogen profile throughout the canopy, Mercado *et al.*, 2009b; Lloyd *et al.*, 2010; Dewar *et al.*, 2012).

Nitrogen limitation and the nitrogen cycle are not yet represented explicitly in HadGEM2-ES but will be in future versions of this earth system model (i.e. UKESM1). Presently, nitrogen allocation at the leaf level ($N_L$) within the canopy is represented via an exponential profile in the land surface code of HadGEM2-ES, that is:

$$N_L = N_{L0}e^{-K_N L}dL \qquad \text{Eq. (1)}$$

where, L is the leaf level Leaf Area Index, $N_{L0}$ is the nitrogen concentration at canopy top (in kgN/kgC) and $K_N$ is a dimensionless constant representing the steepness of the nitrogen profile. A shallow nitrogen profile ($K_N$=0.128) is the

JULES default (Mercado et al 2007) and is assumed in HadGEM2-ES for the Main Experiment. For this sensitivity experiment, we investigate the consequence of assuming a steeper nitrogen profile ($K_N$=0.5). Under these conditions, one might expect lesser light use efficiency under diffuse light conditions as shaded leaves become nitrogen limited (Hikosaka 2014). We re-run the BBAx0, BBAx1 and BBAx2 simulations using the steeper nitrogen profile (labelled BBAx0$_{STEEP\_N}$, BBAx1$_{STEEP\_N}$ and BBAx2$_{STEEP\_N}$ respectively).

To derive a new parameter value of $K_N$ which still provides consistent global NPP fluxes, we repeated the offline analyse described in in Sect. 2.1. We used JULES to perform 1D simulations of a tropical site with varying combination of the $K_N$ and $N_{L0}$ parameters to derive biochemical fluxes (Supplementary Fig. S4b and S4c). The parameter combination were chosen such as the mean canopy carboxylation rate ($V_{cmax,25,C}$) is conservative and remained at the same level as in the main

experiment (i.e. about 27 μmol $CO_2$.m$^{-2}$.s$^{-1}$ for broadleaf trees). Nitrogen allocation being represented by an exponential decay, the mean canopy $V_{cmax,25,C}$ can be calculated as follows:



$$V_{cmax,25,C} = n_e N_{L0} \frac{(1 - e^{-K_N LAI})}{K_N LAI} \qquad \text{Eq. (2)}$$

Where LAI is the Leaf Area Index at canopy level, $n_e$ is a constant that has values of 0.0008 and 0.0004 mol $CO_2$.m$^{-2}$.s$^{-1}$ kgC (kgN)$^{-1}$ for $C_3$ and $C_4$ plants, respectively (Mercado et al 2007).

### 2.3.4 Atmospheric $CO_2$ concentration

It is hypothesised that in a richer $CO_2$ world, rates of photosynthesis will increase and in addition plants could afford reduced stomatal opening to fix the same amount of $CO_2$, resulting in a higher water use efficiency which should further enhance plant productivity – the so-called $CO_2$ fertilization effect (e.g. Keenan *et al.*, 2012). As stated earlier, fires do not only release aerosol particles but also $CO_2$, amongst other gases, which locally increases background $CO_2$ levels (e.g. Wittenberg *et al.*, 1998). Additionally, it is expected that the rise in atmospheric $CO_2$ will continue given current projections of anthropogenic emissions (O'Neil *et al.*, 2016). The details of the $CO_2$ fertilization effect are complex because environmental changes occur simultaneously (e.g. van der Sleen *et al.*, 2015; Zhu *et al.*, 2016). It would be far beyond the scope of this study to fully characterise the $CO_2$ fertilization effect strength in HadGEM2-ES but it is certainly of interest to evaluate if the effect of aerosols on vegetation through alteration of the surface PAR differs when the atmospheric background $CO_2$ is varied. For this experiment, the BBAx0, BBAx1 and BBAx2 simulations are done twice, once with the level of background $CO_2$ increased by +25 ppm globally and once with an increase of +50ppm globally. Increments of +25 and +50ppm should be representative of the $CO_2$ level expected in 12.5 and 25 years respectively if one assumes a 2 ppm/a increase (as supported by the mean rate of $CO_2$ increase measured at Mauna Loa for the period 2000-2010).

### 2.4 A framework to analyse the changes in fast carbon fluxes

As stated previously, aerosols can affect photosynthetic rates through different pathways (e.g. Bonan 2008 and Supplementary Fig. S7). Firstly, by altering the amount of light (the '*reduction in total PAR*') and light quality (the '*change in diffuse fraction*' of PAR). Secondly, aerosols interact with radiation and clouds impacting the climate directly and indirectly which affects the radiative balance therefore the energy budget, forcing the coupled system to adjust to the aerosol perturbations. These adjustments (the '*climate feedback*') can feedback into the calculations of the rate of vegetation biochemical processes – e.g. by altering the surface temperature. A simple theoretical framework can be used to discriminate a fast carbon flux, e.g. NPP, as a function of the '*change in diffuse fraction*', $f_d$, the '*reduction in total PAR*', *TotPAR* and the '*climate feedback*', *clim*, such as NPP($f_d$, *TotPAR*, *clim*). Neglecting the interdependency between the three *terms*, enables the following decomposition:

$$\delta NPP = \frac{\partial NPP}{\partial f_d} \delta f_d + \frac{\partial NPP}{\partial TotPAR} \delta TotPAR + \frac{\partial NPP}{\partial Clim} \delta Clim \qquad \text{Eq. (3)}$$





To evaluate how these three terms contribute individually to the total change in NPP (the '*net impact*'), we have developed three new model diagnostics in HadGEM2-ES. For each model time-step, we diagnose four surface fluxes of PAR which are the Total and Direct PAR, considering or excluding the aerosol radiative effects. This is achieved by calling the radiative transfer routines twice (i.e. a double call) within the same model time-step; i.e. first call with the aerosol radiative effects

considered, and second call assuming 'clean-sky' conditions where the radiative effects of aerosols are not considered (Ghan, 2013). Note that the effect of clouds on the radiative fluxes are always considered during the two calls. The next model iteration (i.e. the prognostic call) always includes the aerosol radiative effects in order to account for their impact on the atmospheric state. That means that the calculation of vegetation processes which occurs after the radiative transfer will always 'see' the climate that has been modified by the aerosols. After the radiative transfer calculations, the four fluxes of

PAR that have been calculated are passed to the physiology routines of JULES to calculate plant productivity. Prior to calculating the biochemical fluxes, we define two values of $f_d$ and *TotPAR* using the four PAR fluxes previously introduced; one that considers the effect of aerosols ($f_{d.aer}$ and *TotPAR$_{.aer}$*) and one that considers 'clean-sky' conditions ($f_{d.clean}$ and *TotPAR$_{.clean}$*).

| | | Aerosol effect on model variables during the triple call: with (.aer) and without (.clean) aerosol effect. | | | | |
|---|---|---|---|---|---|---|
| | | *fd* | *TotPAR* | *clim* | **Biochemical flux diagnostic (e.g. NPP)** | *Comments* |
| Call order of the physiology routines | **#1** | $f_d$.clean | *TotPAR*.clean | *clim*.aer | NPP$_{clim.aer,TotPAR.clean,fd.clean}$ | *NPP of vegetation only experiencing the change in climate* |
| | **#2** | $f_d$.clean | *TotPAR*.aer | *clim*.aer | NPP$_{clim.aer,TotPAR.aer,fd.clean}$ | *#2 minus #1 = impact of change in total amount of PAR* |
| | **#3** | $f_d$.aer | *TotPAR*.aer | *clim*.aer | NPP$_{clim.aer,TotPAR.aer,fd.aer}$ | *#3 minus #2 = impact of change in diffuse fraction of PAR* |

**Table 3 - Model quantities calculated during the triple call of the physiology routines (see text).**

The physiology routines are then called three times (i.e. a triple call, see Table 3) within the same model time-step. On the first call, both the '*reduction in total PAR*' and the '*change in diffuse fraction*' are ignored (i.e. the vegetation only sees the '*climate feedback*'). The biochemical fluxes calculated during this first call are saved in a specific model diagnostic

($NPP^{BBAxx}_{clim.aer,Tot.clean,fd.clean}$). On the second call, the '*reduction in total PAR*' due to aerosols is then considered but the '*change in diffuse fraction*' of PAR is not accounted for and a new set of biochemical fluxes are saved in a specific model diagnostic ($NPP^{BBAxx}_{Clim.aer,Tot.aer,Fd.clean}$). For the last prognostic call, both aerosol effects on '*reduction in total PAR*' and the '*change in diffuse fraction*' are taken into account in the calculation of the biochemical fluxes and saved in a specific model diagnostic ($NPP^{BBAxx}_{Clim.aer,TotPAR.aer,F_d.aer}$).



With these new diagnostics available, we are able to isolate the impacts of '*change in diffuse fraction*', '*reduction in total PAR*' and '*climate feedback*' by comparing model simulations which include or exclude the BBA emissions. For instance, the effect of BBA in the BBAx1 simulation (i.e. the standard emissions scenario) can be expressed as follows:

$$\Delta \overline{NPP}_{net\ impact}^{BBAx1} = \overline{NPP}^{BBAx1} - \overline{NPP}^{BBAx0} = \Delta \overline{NPP}_{f_d}^{BBAx1} + \Delta \overline{NPP}_{TotPar}^{BBAx1} + \Delta \overline{NPP}_{clim}^{BBAx1} \qquad \text{Eq. (4)}$$

5   with,

$$\Delta \overline{NPP}_{F_{dPAR}}^{BBAx1} = \left( \overline{NPP}_{clim.aer,Tot.aer,Fd.aer}^{BBAx1} - \overline{NPP}_{clim.aer,Tot.aer,Fd.clean}^{BBAx1} \right)$$
$$- \left( \overline{NPP}_{clim.aer,Tot.aer,Fd.aer}^{BBAx0} - \overline{NPP}_{clim.aer,Tot.aer,Fd.clean}^{BBAx0} \right) \qquad \text{Eq. (5)}$$

$$\Delta \overline{NPP}_{TotPar}^{BBAx1} = \left( \overline{NPP}_{clim.aer,Tot.aer,Fd.clean}^{BBAx1} - \overline{NPP}_{clim.aer,Tot.clean,Fd.clean}^{BBAx1} \right) \qquad \text{Eq. (6)}$$
$$- \left( \overline{NPP}_{clim.aer,Tot.aer,Fd.clean}^{BBAx0} - \overline{NPP}_{clim.aer,Tot.clean,Fd.clean}^{BBAx0} \right)$$

$$\Delta \overline{NPP}_{clim}^{BBAx1} = \left( \overline{NPP}_{Clim.aer,Tot.clean,Fd.clean}^{BBAx1} \right) - \left( \overline{NPP}_{Clim.aer,Tot.clean,Fd.clean}^{BBAx0} \right) \qquad \text{Eq. (7)}$$

where overbars denote quantities averaged over a time period long enough for vegetation fast responses to adjust to the aerosol effects.

**2.5 Observations used in model evaluation**

We evaluate global fields of simulated GPP and NPP using GPP fields derived by the FLUXCOM project (Tramontana *et al.*, 2016; Jung *et al.*, 2017) and the global annual mean NPP retrievals based on the MODIS MOD17A2 product (Running *et al.*, 1994) (Figs 1a and 1b). The GPP from FLUXCOM is derived from a model that has been trained on observational data so we will refer to this estimate as a 'reconstructed' GPP. In addition, in-situ estimates of NPP from the EMDI project (http://gaim.unh.edu/Structure/Intercomparison/EMDI/) are also presented in the form of overlaid circles depicted in Fig. 1b. Note, simulated values of HadGEM2-ES GPP and NPP used in the comparison with observational data are sampled where the corresponding observationally based dataset contains non-missing data.

The simulated aerosol loading is evaluated against the record of Aerosol Optical Thicknesses (AOT) retrieved from the MODIS instrument measurements on board of the TERRA satellite. The dataset used corresponds to the collection 6.1 monthly mean 1-degree Level 3.0 products that were derived from the MYD06_L2 products for the period extending between 2001 and 2016.



Additional evaluation of the model skill against observations is provided in the supplementary material (Supplementary Fig. S8). This includes comparison of the modelled solar fluxes at the surface against the SSF 1-degree Terra Edition 2.8 product based on the CERES radiation data, and comparison of the modelled surface precipitation against the GPCP version 2.3 product.

## 3 Results

### 3.1 Evaluation

#### 3.1.1 Carbon exchange

Global annual mean GPP and NPP as simulated by HadGEM2-ES with the new representation of canopy light interception are shown in Fig. 1c and 1d. The global GPP modelled by HadGEM2-ES is +115 PgC/yr in the updated version of
HadGEM2-ES and smaller than the estimate of +129 PgC/yr from the FLUXCOM dataset (Fig. 1a) but closer to the reference of +118 PgC/yr cited by Shao *et al.* (2013). The standard configuration of HadGEM2-ES that participated in CMIP5 had a global GPP of the order of +140 PgC/yr for present-day conditions (Shao *et al.*, 2013). Despite the variation between the two reference estimates for the global GPP (i.e. between +118 and 129 TgC/yr), this suggests that the updated version of HadGEM2-ES is able to provide better estimate of the global GPP. The global NPP modelled by HadGEM2-ES is
+54 PgC/yr (Fig. 1d) and in good agreement with the satellite-based estimate of +50 PgC/yr (Fig. 1b) and the "best guess" value of +56 PgC/yr reported by Shao *et al.* (2013). The updated configuration of HadGEM2-ES performs well in mid/high latitudes, particularly against EMDI data (Fig. 1d) but biases still remain in the tropics (Fig. 1f) particularly over South America in areas dominated by C3 grass (Supplementary Fig. S5).

#### 3.1.2 Biomass Burning Aerosols

Biomass burning is highly variable from year to year. This can be readily observed by monitoring the aerosol optical thickness (AOT), a proxy for the amount of aerosol particles present in the atmosphere. Figure 2a shows the averaged AOT retrieved at 550nm for the months July-August-September (JAS) between 2001 and 2016 by the MODIS instrument on board of the TERRA satellite. Although most of man-made fires occur in the so-called arc of deforestation on the edge of the rainforest (Cochrane 2003), the hot spot of high AOT (>0.6) is actually observed over the Rondonia state (Brazil) near the
Bolivian border. This hotspot can be explained by *i*) the action of the large-scale atmospheric circulation that recirculates aerosols over South America, and *ii*) the contribution of natural fires that occur concomitantly with fires of anthropogenic origin. Figure 2c provides more detail on the AOT variability by showing the seasonal cycle calculated over central Amazon (i.e. the region encapsulated in the red box shown in Fig. 2a using the multiyear data record from MODIS). The mean seasonal cycle for the period 2001-2016 is represented by the thick black line, while the individual years contributing to the multi-year average are represented by the red-dashed lines. Despite year-to-year variability, AOT is found to peak in





September over this region that is, at the expected peak of the fire season, supporting that BBA are the dominant component of the total aerosol loading during that period.

The AOT modelled by HadGEM2-ES in the simulation that assumes standard BBA emission (i.e. the BBAx1 simulation) is

in overall good agreement with the MODIS observations for the JAS period (Fig. 2a, 2b, Johnson *et al.*, 2016). However, the AOT at the peak of the fire season (i.e. in September) is underestimated (Fig. 2d). In contrast, the modelled AOT for September in the BBAx2 simulation is in better agreement with the satellite retrievals. We will therefore consider in the remainder of this paper that the combination of BBAx1 and BBAx2 scenarios are representative of present-day levels of BBA and will use them to discuss the effects of BBA on the rainforest productivity. There is huge variation in the inter-

annual variation in the magnitude of the AOT (Fig. 2c), which justifies the upper bound for our simulation scenarios; the simulations BBAx0.5 and BBAx4 will be considered as representative of emissions for years with low and high fire activity, respectively (Fig. 2c). These simulations will provide a lower, respectively upper, estimate of the BBA impact on vegetation.

### 3.1.3 Surface radiation

Figure 3 illustrates the impact of BBA on the radiative fluxes in the HadGEM2-ES simulations. The seasonal cycle of the

Total PAR (*TotPAR*) shows a strong decrease during the whole dry season with the strongest reduction occurring in August/September. The reduction in *TotPAR* is in the range of [-18.0 ; -7.5] W/m$^2$ (i.e. [-14.0 ; -5.5] %) in the BBAx1 and BBAx2 experiments, respectively (Fig. 3a and 3b). For the most extreme emission scenarios (BBAx4), the reduction in *TotPAR* is as high as -30 W/m$^2$ or -25 % in August. Conversely, the diffuse component of PAR (*DiffPAR*) increases with aerosols as expected from the theory of light scattering (Fig. 3c and 3d). The diffuse PAR reaching the top of the canopy is

increased by approximately [+6.0 ; +12.0] W/m$^2$ (i.e. approximately [+14.0 ; +31.0] %) during August/September in the BBAx1 and BBAx2 simulations (Fig. 3c and 3d). Overall this leads to an increase in the diffuse fraction of PAR (i.e. $f_d$) of [+20.0 ; +55.0] % (Fig. 3e and 3f).

An alternative representation of the impact of BBA on the radiative fluxes is depicted in Fig. 4 for August and September.

Here, the composite plot is constructed using the four simulations that include BBA emissions to calculate the *TotPAR* (Fig. 4a), *DiffPAR* (Fig. 4b) and $f_d$ (Fig. 4c) at the surface as a function of the total AOT (i.e. BBA + background aerosols). The composite was constructed by first averaging each simulation over time to create climatologies for the specific months, then all pixels contained in the domain of analysis were sampled to construct the scatterplots of the desired quantities. It is important to note that radiative quantities were sampled for the full sky grid-box and that no conditional sampling was

applied *a priori*, therefore cloud effects are implicitly accounted for in these statistics. Subsequently, further averaging of the data into 30 bins of AOT (respectively, $f_d$ for Fig. 4d) was applied to smooth the signal. Figure 4a shows the expected monotonic decrease in *TotPAR* with AOT. Concomitantly, the *DiffPAR* (Fig 5b) increases with AOT up to values of around 1.75 and decreases for higher AOTs. This illustrates that increasing AOT could only increase the amount of diffuse light





reaching the surface up to a point; above this point, the effect of the attenuation of *TotPAR* dominates. This AOT threshold around 1.75 maximises the amount of diffuse radiation reaching the canopy top. However, as it will be detailed in following sections, this threshold does not correspond to maximum effect of aerosols on vegetation productivity.

### 3.2 The '*net impact*' of BBA on forest productivity

Fig 5d represents NPP as a function of $f_d$ for the months of August and September in the same way as the surface radiative fluxes against AOT are depicted (Fig 5a, b, c). This shows that NPP is likely to reach an optimum when $f_d$ approximately equals to 52-56%. The existence of an optimum $f_d$ that would maximise carbon sequestration is consistent with findings reported in past modelling studies (e.g. Knohl and Baldocchi 2008; Mercado *et al.*, 2009; Pedruzo-Bagazgoitia *et al.*, 2017; Yue *et al.*, 2017a). Such an optimum however, depends strongly on factors such as the vegetation canopy architecture

environmental conditions, solar zenith angle or the optical properties of the scattering particles. The fact that an optimum diffuse fraction emerges is consistent with our understanding of the DFE mechanism. When $f_d$ is lower than the optimum, an increase in the amount of diffuse radiation increases carbon assimilation because a larger area of shaded leaves become photosynthetically active. For $f_d$ beyond the optimum, the effect of the attenuation of *TotPAR* dominates and sunlit leaves are no longer light saturated, resulting in an overall decrease in biochemical fluxes at the canopy level with further increase in $f_d$.

Figure 4c could be used to infer an AOT for which $f_d$ is getting close to the optimum value of 0.55 (Fig. 4d). This would approximately occur at an AOT of ~0.9-1(Fig. 4c). However, we do not observe that the highest NPP enhancement occurs around these values of AOT in our simulations (see Sect. 3.3). This can be understood as a consequence of equifinality, because both the effects of clouds and the effects of aerosols on radiation occur concomitantly. There are then many possible

combinations of cloud and aerosol scenarios that could create optimum conditions maximising the DFE. It would be possible to disentangle the effect of BBA from the effect of clouds on carbon sequestration by either screening out cloudy scenes or diagnosing the biochemical fluxes in the clear-sky portion of the model grid-boxes, providing a mean to quantify the maximum potential impact of BBA on carbon sequestration. This approach was used by Moreira *et al.* (2017) to conclude that BBA could increase the GPP of the Amazon forest by up to 27%. While this study is insightful, our aims here are

different as we seek to understand the impact of BBA while considering the system-wide behaviour, that is including the effects of both aerosols and clouds. This alternative approach was used by Yue et *al.* (2017a) to analyse aerosol impacts on vegetation over China and show that clouds are a dominant feature controlling the diffuse fraction of radiation which modulates the diffuse fertilisation effect from aerosols (Yue et *al.* 2017a, Fig. 5 therein). In Sect. 3.3, we will show that similar conclusions could be drawn over South America.

Despite cloudiness affecting how much aerosols can interact with radiation, we notice that NPP is enhanced in the central part of the Amazon when BBA emissions are increased (Fig. 5). The most statistically significant enhancement of the NPP, which is depicted by the stippling in Fig. 5, occurs during August, in phase with the period when the radiative impacts of





BBA are the most pronounced in the model simulations (Fig. 3, Sect. 3.1.3). Although the simulated AOTs are of similar magnitude during September, NPP enhancement is not as robust as in August (i.e. there is a less statistically significant signal in the NPP anomalies). This can partially be explained by the fact that plant productivity simulated by HadGEM2-ES reaches a minimum in September (Supplementary Fig. S8a and S8b). As a result, the vegetation is less active in September

and the potential impact of the BBA perturbation is reduced.

Overall, based on the BBAx1 and BBAx2 simulations, we estimate that BBA increase NPP by about +80 to +105 TgC/yr, or 1.9 to 2.7% (Fig. 6b and 6c) over the domain of analysis. This estimate of the enhancement in carbon uptake is remarkably similar to the estimate provided by Rap *et al*. (2015) who found that Amazonian fires increase NPP by 1.4 - 2.8% corresponding to an increase of +78 to +156 TgC/yr. This is encouraging as the authors used the stand-alone version of

JULES (i.e. the land surface component in the HadGEM family of models). However, as it will be discussed in Sect. 3.3 and Sect. 4.2, we attribute the enhancement in carbon sequestration to different mechanisms. The Rap *et al.* (2015) study used a combination of offline models which do not account for climatic adjustment to the aerosol radiative perturbation. This supports that the increase in modelled NPP results from DFE in their simulations. Conversely, we will show (Sect. 3.3) that DFE is negligible over the region considered in our model simulations but the overall aerosol impacts on vegetation remains

significant thanks to the contribution of *climate feedbacks* that are experienced by the vegetation.

### 3.3 Disentangling the impact of radiation changes from those of climate adjustments.

We have quantified the '*net impact*' of BBA on NPP in the previous section. Following the framework described in Sect. 2.4, we now address separately the individual contribution from the '*change in diffuse fraction*', $f_d$, the '*reduction in total PAR*', *TotPAR* and the '*climate feedbacks*' to the BBA '*net impact*' on vegetation productivity. Figure 7 shows the seasonal

cycle of NPP anomalies averaged over the domain of analysis (left axis) and the corresponding accumulated anomalies (right axis) for the four simulations with varying BBA emissions. The increase in NPP due to the '*change in diffuse fraction*' is unambiguous (Fig. 7a), corresponding to an enhancement in plants net carbon uptake of +65 to +110 TgC/yr in the BBAx1 and BBAx2 simulations, respectively. As expected, the '*reduction in total PAR*' has the opposite effect and systematically decreases NPP (Fig. 7b) with increasing negative NPP anomalies. This corresponds to a reduction in plant net carbon uptake

of -52 to -105 TgC/yr in the BBAx1 and BBAx2 simulations, respectively. The combination of the '*change in diffuse fraction*' and the '*reduction in total PAR*' effects represents the DFE. We estimate that the DFE from BBA increases the vegetation NPP by +13 and +5 TgC/yr in the BBAx1 and BBAx2 simulations, respectively.

The impact of BBA on NPP via the DFE is in stark contrast with the increase in forest productivity which we have discussed

in the previous Sect. 3.2 describing the '*net impact*' of BBA (+80 to +105 TgC/yr for the BBAx1 and BBAx2 simulations respectively). This would indicate that in our simulations the net impact of BBA on forest productivity is not mostly due to



the DFE. Figure 7c shows that the '*climate feedback*' term is actually the dominant contribution and systematically increases NPP, with an enhancement of +67 to +100 TgC/yr in the BBAx1 and BBAx2 simulations, respectively.

It is worth mentioning that the maximum impact of the '*change in diffuse fraction*' occurs during August in the BBAx4
simulation which increases the NPP by +41 TgC/m. The corresponding impact of the '*reduction in total PAR*' decreases NPP by -66 TgC/m. This illustrates that for a year with intense burning, the system actually seems to shift past the point where the balance between the total and the diffuse PAR does not increase the efficiency of photosynthesis anymore (i.e. BBA DFE leads to reduction of -42 TgC/yr on an annual basis for the BBAx4 scenario). Interestingly, in this simulation, despite the negative impact on NPP from DFE, we note that the impact of '*climate feedback*' is much larger (+194 TgC/yr),
resulting in the '*net impact*' of BBA on the vegetation to be overall positive (+ 151 TgC/yr).

To compare the relative contribution of the DFE (i.e. '*change in diffuse fraction*' plus '*reduction in total PAR*') and the '*climate feedbacks*' on vegetation NPP as the atmospheric aerosol content ramps up, Fig. 8a depicts the relative change in NPP (%) as a function of AOT for the month of August. This NPP change is further decomposed into individual
contributions from: the '*change in diffuse fraction*' (blue solid line), the '*reduction in total PAR*' (red solid line), the DFE (green solid line), the '*climate feedback*' (yellow solid line) and the '*net impact*' (black solid line). The resulting attribution plot shown in Fig. 8a was constructed in the same way as Fig. 4 (see Sect. 3.1), i.e. by first averaging each simulation over time, then sample the NPP changes associated with each of the three terms in all the model grid-boxes from the domain of analysis, and finally aggregating the sampled quantities into 30 bins of AOT.

Overall, it is clear from Fig. 8a that BBAs enhance NPP across the entire range of AOT considered here (i.e. the black solid line representing the '*net impact*' of BBA is strictly positive) which is consistent with the geographic distribution of anomalies displayed on Fig. 5. The impact of the '*change in diffuse fraction*' and the '*reduction in total PAR*', respectively, consistently increases and decreases vegetation NPP, respectively, as discussed in the previous paragraph. However, the
impact of DFE from the BBA (represented by the green solid line in Fig. 8a), changes its sign around AOT of ~0.9. At lower AOTs DFE from BBA contributes to an increase in NPP, whereas at higher AOTs it has the opposite effect. To help visualize the transition in the DFE regime, an enlarged Sect. of the plot from Fig. 8a depicting only the DFE contribution is shown in Fig. 8b. A limited AOT range from 0.0 to 1.5 is shown for the month of August (green solid line) and the month of September (blue solid line). It is interesting to note that the AOT optimum occurs at smaller AOTs in September as
compared to August.

As discussed in Sect. 3.1, changes in NPP due to DFE from BBA alone are calculated under all sky-conditions which also account for cloud radiative effects. A plausible explanation for the observed reduction in the range of AOT creating a positive DFE would be that cloudiness increases over the analysed model domain between August and September (see





Supplementary Fig. S10) as the regional climate progresses towards the wet season. This is supported by the increase in $f_d$ between August and September in the simulation that excludes BBA (i.e. black solid line in Fig. 3c). These results are consistent with those of Yue *et al.* (2017a) who discussed how the impact of anthropogenic aerosols DFE over China vary depending on the cloud cover which allows for smaller or larger perturbations in the radiative balance for the same

atmospheric aerosol loading.

## 3.4 Sensitivity experiments

Here, we present the results from the four additional sensitivity experiments described in Sect. 2.3. These experiments were designed to further elucidate the role BBA play in vegetation productivity while changing some of the underlying assumptions in the previous experiments which relate to *i)* aerosol optical properties, *ii)* aerosol-cloud interactions, *iii)* the

canopy nitrogen profile and *iv)* atmospheric carbon dioxide concentration. Figure 9a shows a box-and whisker plot of NPP averaged over central Amazonia during August for all BBAx0, BBAx1 and BBAx2 simulations from the main experiment (those analysed in the Sect. 3) and from the four additional sensitivity experiments. The mean changes in NPP due to biomass burning aerosols are shown in Fig. 9b.

The results can be summarized as follows:

- *Aerosol optical properties (experiments DIFF_OP and ABS_OP)* – The optical properties of BBA have been altered in order to make the biomass burning aerosols more (*ABS_OP*) or less scattering (*DIFF_OP,* i.e. we modified the SSA, Supplementary Fig. S6a and S6b). The mass specific extinction is invariant (see Sect. 2.3.3) which implies that for the same AOT, the direct radiation reaching the surface is also independent of the aerosol scattering/absorbing

efficiency assumptions (Supplementary Fig. S6c). More scattering or absorbing BBA, respectively, increase/decrease the diffuse fraction of solar radiation reaching the surface (Supplementary Fig. S6c). As a result, scattering BBA should produce a stronger DFE and absorbing BBA should analogously produce a weaker DFE. However, we do not observe a significant change in the modelled BBA impact on vegetation productivity for the varying BBA scattering/absorbing assumptions (Fig. 9b) because as discussed in Sect. 3.3, the DFE from present-day BBA is very

small for this model in this region. Therefore, altering the ratio of diffuse fraction reaching the ground via the aerosol optical properties, that is modulating the magnitude of the DFE, do not have a significant impact on vegetation productivity.

- *Aerosol-Cloud-Interactions (ACI – experiments 1ˢᵗAIE and NoAIE)* – We have emphasised the potential role of clouds in Sect. 3.3. One could expect that increasing aerosol emissions which provide the necessary CCNs will increase

cloud droplet numbers and reduce their sizes. The reduction in droplet size leads to cloud brightening (1ˢᵗ AIE) and possibly cloud amount (2ⁿᵈ AIE), which could eventually alter the surface radiation balance. We note that the impact of BBA on NPP is of similar magnitude in the main experiment and in the experiments without aerosol-cloud



interactions (Fig. 9b) – i.e. neglecting ACI do not change the impact of BBA on vegetation productivity over the region considered. A possible explanation can be found in the type of the clouds that predominates in this region. We note that most of the precipitation in HadGEM2-ES stems from convective clouds. Aerosols are only coupled to the large-scale precipitation scheme in HadGEM2-ES (i.e. aerosols can only alter the properties of stratiform clouds). The absence of any impact from ACI over this region is then to be expected. Whether or not ACI can affect vegetation productivity remains a research topic for future studies and these should focus on regions where aerosols and clouds are likely to interact as a consequence of the cloud representation in the models (e.g. Chameides *et al.*, 1999). Alternatively, the ACI effects in the cloud representation should be revisited and improved in the models (Malavelle *et al.*, 2017).

- *Canopy nitrogen profile* (experiment STEEP_N): We modified the shape of the nitrogen profile for the modelled canopy to represent a steeper decrease in leaf nitrogen content (Sect. 2.3.4). The available nitrogen to leaves decreases from the canopy top downwards. This change in leaf nitrogen allocation means that sunlit leaves have access to more resources, whereas shaded leaves tend to be more nitrogen limited (Hikosaka *et al.*, 2014). Despite this modification in nitrogen availability, we do not observe a significant change in the modelled BBA impact on vegetation productivity. The reasons for this absence of sensitivity to nitrogen availability are similar as in the experiments testing the role of aerosol optical properties, i.e. the DFE from BBA is already too small to have a discernible impact and reducing the allocated nitrogen in the shaded portion of the canopy only reduces its impact more.

- *Atmospheric CO$_2$ concentration* (experiments 25ppm and +50ppm). While increasing atmospheric CO$_2$ concentration leads to an unambiguous increase in NPP (Fig. 9a), the BBA impact is of similar magnitude as in the main experiments (Fig. 9b). It may appear that the impact of BBA is somehow reduced in the +25ppm case compared to the main experiment and the +50ppm experiment. However, the level of model internal variability in NPP is too pronounced (Fig. 9a) to draw robust conclusions on the impact of a variation in CO$_2$ on the BBA-induced DFE. Note that the atmospheric CO$_2$ concentration increased globally. It was also allowed to affect the radiative balance resulting in a warming climate in these two experiments. Potentially, this could increase the model's internal variability further. If one were to repeat these experiments, only the leaf-internal CO$_2$ concentration should be increased to avoid additional statistical noise produced in the warming climate.

## 4. Concluding remarks

From our model experiments we concluded that the diffuse PAR fertilisation effect from biomass burning aerosols in HadGEM2-ES (Sect. 3.3) is comparatively modest amounting to between +13 and +5 TgC/yr based on the result from the simulations BBAx1 and BBAx2. This may seem in odds with the +78 to +156 TgC/yr estimate (assuming respectively standard BBA emissions and 3 times the standard BBA emissions) reported by Rap *et al.* (2015), who used the JULES land surface model in an offline framework specifically designed to assess the DFE of biomass burning aerosols. Some



differences between the two studies that could explain the apparent differences are obvious, such as for instance the fact that we are not reporting estimates for the BBA impact over the same area (i.e. our domain is smaller) or that we did not use the same aerosol properties or emission inventories. We recalculated the impact from biomass burning aerosols in our simulations over a larger domain that approximately matches the area considered by Rap *et al.* (2015). In this situation, we

found that the net increase in NPP is about +145 to +148 TgC/yr for the BBAx1 and BBAx2 respectively, of which only +15 to +5 TgC/yr are attributable to the DFE. This confirms that the magnitude of the DFE from BBA effect is small increasing plant productivity in our simulations over the Amazon forest.

Biases in the cloud amount which is inherent of coarse model parameterisations may affect the surface radiation and impact
the magnitude of the DFE from biomass burning aerosols (and indeed all aerosols). Those uncertainties can partially be contained using an offline framework where the state of the model can be forced closer to the distribution of input observations. However, in this approach, internal consistency is lost by not allowing variability within non-linear relationships (e.g. how cloudiness is changed due to aerosol-radiation interactions, how plant dark respiration is changed due to the surface cooling). This then poses a problem and a risk of overestimating the response of a component (e.g. vegetation
productivity) to a perturbation such as those introduced by aerosols. By including more complexity in a coupled framework as in the present study, we believe that our estimate of the DFE is more consistent, albeit being low due to possible uncertainties/biases, and we would argue that earlier estimates of the DFE from BBA in this region (Rap *et al.*, 2015) are probably on the high end. Nonetheless, despite showing that the DFE from BBA is not an efficient mechanism in our simulations over this region, we have demonstrated a pathway where BBA can significantly influence vegetation
productivity. We assessed this pathway by calculating the term representing biomass burning aerosol '*climate impact*' on vegetation which represents the rapid adjustments of land-surface climate to aerosol radiation perturbation. We estimated this term to be about +67 to +100 TgC/yr over the domain analysed in this study in the BBAx1 and BBAx2 simulations, respectively. This is a novel contribution which could not be accounted for in an offline modelling framework and has therefore not been properly assessed in past studies. This term is non-negligible, and potentially in line with the impact from
other biomass burning by-products.

We can now proceed to compare the impact of BBA over Amazonia with the effect of $O_3$ on the vegetation that is produced from $O_3$-precursors emitted by forest and grassland fires. Although Pacifico *et al.* (2015) reported the changes in GPP, their results can be directly compared to the changes in NPP derived from our simulations because the effects of BBA in
HadGEM2-ES are predominantly affecting the GPP whereas the impact on plant respiration is of second order over this region of the world under present-day climate (Supplementary Fig. S9). Using the same modelling set-up as in the present study, Pacifico *et al.* (2015) estimated that present-day $O_3$ produced from precursors emitted by forest and grassland fires in the Amazon region reduces the vegetation GPP by approximately -230 TgC/yr over the same region that has been analysed in this study. This is about two times, but of opposite sign, the magnitude of the '*net impact*' of BBA estimated in this study





(i.e. +80 to +105 TgC/yr for the BBAx1 and BBAx2 scenarios) which includes the *'climate feedbacks'*. However, it is important to emphasize that the result from Pacifico *et al.* (2015) is based on an approach of modelling the $O_3$ effects on photosynthesis that includes a "high" and "low" parameterization for each plant functional type to represent species more sensitive and less sensitive to $O_3$ effects. The -230 TgC/yr decrease in GPP reported there is based on the "high" sensitivity

mode to establish the maximum response. It is also worth noting that due to a lack of knowledge and data on the impacts of $O_3$ on tropical vegetation, the $O_3$ damage parameterization in the work by Pacifico *et al.* (2015) was derived from data from the temperate and boreal regions. As discussed in the previous paragraph, the BBA-induced DFE is small in our simulations and if an upper estimate of the BBA were to be considered, it is then possible to argue that BBA have the potential to virtually counteract the $O_3$ leaf damage resulting from biomass burning in the area. However, while the biomass burning and

$O_3$ impacts are potentially of the same magnitude but of opposite sign they are not geographically collocated. This means that BBA and $O_3$ do not necessary affect the same regions of the Amazon rainforest. As reported in Pacifico *et al.* (2015) $O_3$ tends to show its highest concentrations upwind of the fires which is located over dense areas of broadleaf trees in the model. In contrast to this, the highest AOT from BBA is found downwind of the fires and located over predominantly grassland areas. Future research aimed at assessing the overall net impact of forest and grassland fires on ecosystems through the $O_3$

and DFE effects should therefore consider modelling the two effects simultaneously in a fully coupled framework.

We showed in Sect. 3.3 that the impact of BBA on vegetation over the Amazon rainforest is dominated by the contribution of the term we have referred as '*climate feedbacks*'. The (bio)physical mechanisms involved behind this term are numerous, and it is beyond the scope of this paper to completely untangle and quantity them. Future work should seek to understand how aerosol can benefit vegetation productivity when the DFE does not suffice to explain the increase in vegetation NPP.

Two working hypotheses for making progress are proposed; first we have noted that BBA are capable of cooling surface temperatures significantly which potentially reduces evapotranspiration (ET) and consequently water stress due to a low soil moisture content (Supplementary Fig. S11a and S11b). Remarkably, the canopy-level Water Use Efficiency (WUE=GPP/ET) is significantly enhanced under higher BBA conditions (Supplementary Fig. S11d). Given the modest increase in GPP reported earlier, it probably implies that the decline in ET was steeper than the increase in GPP and this

would suggest that vegetation is able to balance water loss and carbon uptake with increasing aerosol concentrations.

Secondly, we suggest that future studies put an emphasis on how BBA can modify the biotic (e.g. rate of carboxylation of the Rubisco enzyme, $V_{cmax,}$ leaf temperature) and abiotic factors (air temperature, Vapour Pressure Deficit, PAR, leaf surface temperature, $CO_2$ concentration and air pressure) which control the vegetation response (Lloyd et al 2008; Wang *et al.*, 2018). We found that the cooling effect of BBA (Supplementary Fig. S12a) actually reduces the leaf temperature beyond

$V_{c,max}$ temperature optimum which works to reduce plant productivity (Supplementary Fig. S12c). But the aerosol cooling also lowers the VPD (vapour pressure deficit) which can stimulate stomatal conductance and thus enhance canopy photosynthesis (Supplementary Fig. S12b). The antagonistic effects from VPD and $V_{cmax}$ changes are particularly relevant to




the sunlit leaves as this population of leaves is mostly rubisco limited in our modelling framework (not shown). Assessing the role of these eco-physiological mechanisms is critical for developing a better understanding of the ecosystem-climate feedbacks which control the carbon flux from the atmosphere to the land-surface and more attention should be paid to this issue. Further research on the ecosystem-climate feedback will also contribute significantly to understand the complex

relationships between aerosols and ecosystems (e.g. Schiferl and Heald, 2018).

## 5 Summary

Intense biomass burning events happen regularly in the vicinity of the Amazon rainforest during the dry season (~August-September), releasing huge amounts of trace gases, aerosols and ozone and aerosol precursors. This potentially leads to very large interactions between chemistry, aerosol, clouds, radiation and the ecosystems.

In this study, we have investigated the impact of biomass burning aerosols (BBA) emissions under present-day conditions on the photosynthesis rate and net primary productivity (NPP) of the Amazon rainforest. Aerosol impacts have many impacts that could influence the ecosystems on a regional scale. Amongst these, light scattering from aerosols is often expected to promote more efficient use of radiation by vegetation through the so-called Diffuse PAR Fertilisation Effect (DFE). To understand the potential impact of BBA in this region, we have implemented an updated representation of plant

photosynthesis and carbon uptake that is sensitive to diffuse light radiation in the UK Met Office HadGEM2-ES earth system model.

Overall, our simulations indicate that the '*net impact*' of BBA increases vegetation NPP by +80 to +105 TgC/yr over the central Amazon basin (Sect. 3.2). For the first time we have separated the contribution from the individual radiative and climatic processes that contribute to our estimate of the BBA '*net impact*' on the vegetation. We found that the increase in

diffuse PAR i) stimulates photosynthesis in the shaded part of the canopy and increases NPP by +65 to +110 TgC/yr in our simulations ii) reduces leaf temperature and together with other climatic feedbacks increasing NPP by +67 to +100 TgC/yr and iii) reduces the total amount of radiation therefore decreasing NPP by -52 to -105 TgC/yr, with an overall impact of BBA beneficial for the vegetation.

In our simulations, the DFE from BBA aerosols is small over the analysis region. Our results do not imply however, that

diffuse light is not effective at stimulating vegetation productivity, rather that is only one of a number of responses to a perturbation in the flux of BBA to the atmosphere. We have discussed some possible reasons why the DFE from BBA appears to be weak in our modelling study (Sect. 3.3 and 4.2). Aerosols are not the only light scatterers present in the atmosphere; clouds too, strongly modify the amount and quality of the radiation reaching the surface. Aerosol-induced DFE impacts may then also depend on cloud cover which allows for smaller or larger radiative perturbations for the same level of

aerosols (e.g. Cohan *et al.*, 2002; Yue *et al.*, 2017a). Future studies seeking to investigate the DFE of aerosols should





therefore critically asses the role played by clouds in providing the baseline diffuse light conditions at the surface before assessing the perturbation associated with aerosol emissions.

The novel result from this study is showing that aerosol impacts on vegetation can be significant thanks to the contribution of the *climate feedbacks* which are the result from the system adjustment to the aerosol perturbations which ultimately affect

vegetation productivity. Those impacts can only be captured when considering the BBA effects in a fully coupled modelling framework. Because the aerosol cooling at the surface has a strong effect on biotic and abiotic processes which control the vegetation response (Wang *et al.*, 2018), future work should invest effort into understanding how the effects of BBA, and other aerosols more generally, can affect the surface energy budget which preconditions photosynthetic activity. This step will certainly become even more relevant as advances in the representation of vegetation physiology and phenology in ESMs

are made (e.g. increasing plant functional types or improving vegetation traits), which would likely lead to different vegetation sensitivities to aerosol effects.

Our modelling study specifically aimed at quantifying the changes in the fast ecosystem responses (e.g. NPP/GPP) in response to the effects of BBA. Because the design of our simulations prevents the slow carbon pools to adjust, we cannot investigate how BBA affects carbon allocation and the potential impact it could have on vegetation structure and dynamics.

More research is required to investigate how the impacts of BBA, and indeed all aerosols, on light and on the surface energy budget may alter the onset and shutdown dates of photosynthesis, growing season length and the canopy structure that provide a feedback to vegetation productivity (Yue *et al.*, 2015). Such feedbacks could become even more relevant under a future warmer climate as anthropogenic aerosol emissions are expected to decrease while vegetation will continue to experience more and more stressful climatic conditions (e.g. Schiferl and Heald, 2018).

**Code availability**

HadGEM2-ES, JULES and SOCRATES codes are available from https://code.metoffice.gov.uk/ for registered users. To register for an account, users should contact their local institutional sponsor. If in doubt, please contact Scientific_Partnerships@metoffice.gov.uk for advice stating your affiliate institution and your reason for wanting access.

**Data availability**

The MODIS cloud and aerosol products (http://dx.doi. org/10.5067/MODIS/MYD06_L2.006) are available from https://ladsweb.modaps.eosdis.nasa.gov/. The CERES radiation data are from SSF 1-degree Terra Edition 2.8, available from https://ceres.larc.nasa.gov/order_data.php. GPCP version 2.3 combined precipitation datasets are available from https://www.esrl.noaa.gov/psd/data/gridded/data.gpcp.html. The FLUXCOM data are available from the Data Portal of the Max Planck Institute for Biogeochemistry https://www.bgc-jena.mpg.de/geodb/projects/Home.php. The CRU datasets are





available from http://www.cru.uea.ac.uk/data. MODIS MOD17A2 NPP product was accessed from
https://neo.sci.gsfc.nasa.gov/view.php?datasetId=MOD17A2_M_PSN. The EMDI data are accessible from
http://gaim.unh.edu/Structure/Intercomparison/EMDI/.

**Author contributions**

F.F.M. (text, implementation of parameterisation in the HadGEM2-ES model, setting up the HadGEM2-ES, JULES and
SOCRATES simulations, processing and analysis of the model outputs), G.A.F., L.M.M., S.S. & N.B. (text, model
development and implementation of parameterisation in JULES and HadGEM2-ES), P.A. & J.M.H. (text, project
coordination of SAMBBA and aircraft deployement).

**Competing interests**

The authors declare that they have no conflict of interest.

**Special issue statement**

will be included by Copernicus.

**Acknowledgments**

This work was funded by the Natural Environment Research Council (NERC) South AMerican Biomass Burning Analysis
(SAMBBA) project grant code NE/J010057/1. G.A.F. was supported by the Joint UK BEIS/Defra Met Office Hadley Centre
Climate Programme (GA01101) and the European Union's Horizon 2020 Research and Innovation Programme under grant
agreement no. 641816 (CRESCENDO). F.F.M. & J.M.H. were part-funded by the NERC SWAAMI grant NE/L013886/1.
L.M and N.B. were partly supported by the UK Natural Environment Research Council through The UK Earth System
Modelling Project (UKESM, Grant No. NE/N017951/1). P.A. acknowledges FAPESP (Fundacao de Amaparo a Pesquisa do
Estado de Sao Paulo) projects 2017-17047-0, 2013/05014-0 and 2012/14437-9 and acknowledges the support from LBA
Program that is managed by INPA (The Brazilian National Institute for Amazonian Research).



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



**List of figures:**

Figure 8: Showing on a) the relative changes in NPP ($\Delta NPP_{Net}$, in grey), the relative changes in NPP due to the '*change in diffuse fraction*' ($\Delta NPP_{Frac\ Diff}$, in blue), '*reduction in total PAR*' ($\Delta NPP_{TOTPAR}$, in red), the sum of '*change in diffuse*







**Figure 1: Global annual estimates of Gross Primary Productivity (GPP, left) and Net Primary Productivity (NPP, right). Observationally based estimates from FLUXCOM MTE analysis (a), MODIS MOD17A2 (b), and HadGEM2-ES (c, d). Zonal mean are shown in e) and f). The circles on the NPP maps (b, d) represent in-situ estimates from the EDMI project (reference).**



**Figure 2: Multi-annual mean for the June-July-August season (JAS) of the Aerosol Optical Thickness (AOT) at 550 nm (a, b) and the seasonal cycle (c, d) of the AOT calculated over the domain highlighted in red for the MODIS TERRA retrieval (a, c) and the HadGEM2-ES model (b, d). The MODIS seasonal cycle (c) shows the multi-year (2001-2016) mean in black line and the individual years are overlaid in red dashed lines. The seasonal cycle for HadGEM2-ES (d) shows the 30 years mean for the 5 experiments with varying biomass burning emissions (see text, section 2.2).**





**Figure 3: Modelled seasonal cycle from HadGEM2-ES for the Total PAR (a, b), the diffuse PAR (c, d) and fraction of radiation that is diffuse (e, f) for the five BBA emissions experiments. Absolute values (a, c, e), and relative anomalies (b, d, f) w/r to experiment BBAx0 (i.e. no biomass burning aerosols). Transparent coloured areas in (a, c, e) corresponds to +/- standard deviation. Dashed lines are the multi-year annual means.**



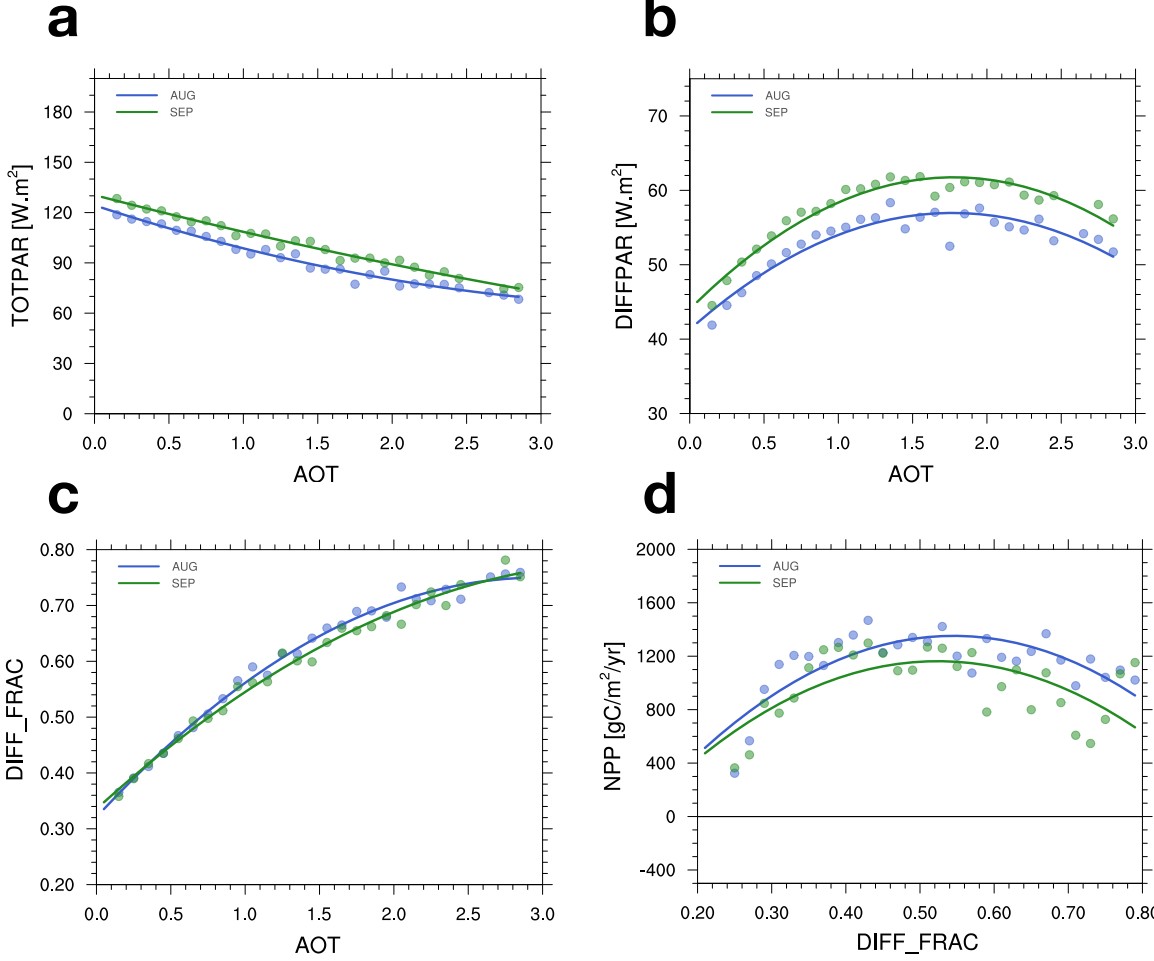

**Figure 4: Showing the Total PAR (TOTPAR, a), the diffuse PAR (DIFFPAR, b) and the fraction of PAR that is diffused (DIFF_FRAC, c) reaching the surface, versus the total Aerosol Optical Thickness (AOT) at 550nm and the Net Primary Productivity (NPP, d) against the fraction of PAR that is diffused.  Circles represent the binned data from the HadGEM2-ES simulations while plain lines are the corresponding 2$^{nd}$ order polynomial fits. Prior to binning, data were first collected at all grid cells in the Amazon region (i.e. the red box region on Fig. 3) for all five BBA emission experiments. We then aggregate all grid cells into 30 AOT bins ranging from 0 to 3 at an interval of 0.1. In each bin, we calculate average AOT and corresponding TOTPAR / DIFFPAR / DIFF_FRAC/ (respectively, we calculate average DIFF_FRAC and corresponding NPP on fig. 5d).**





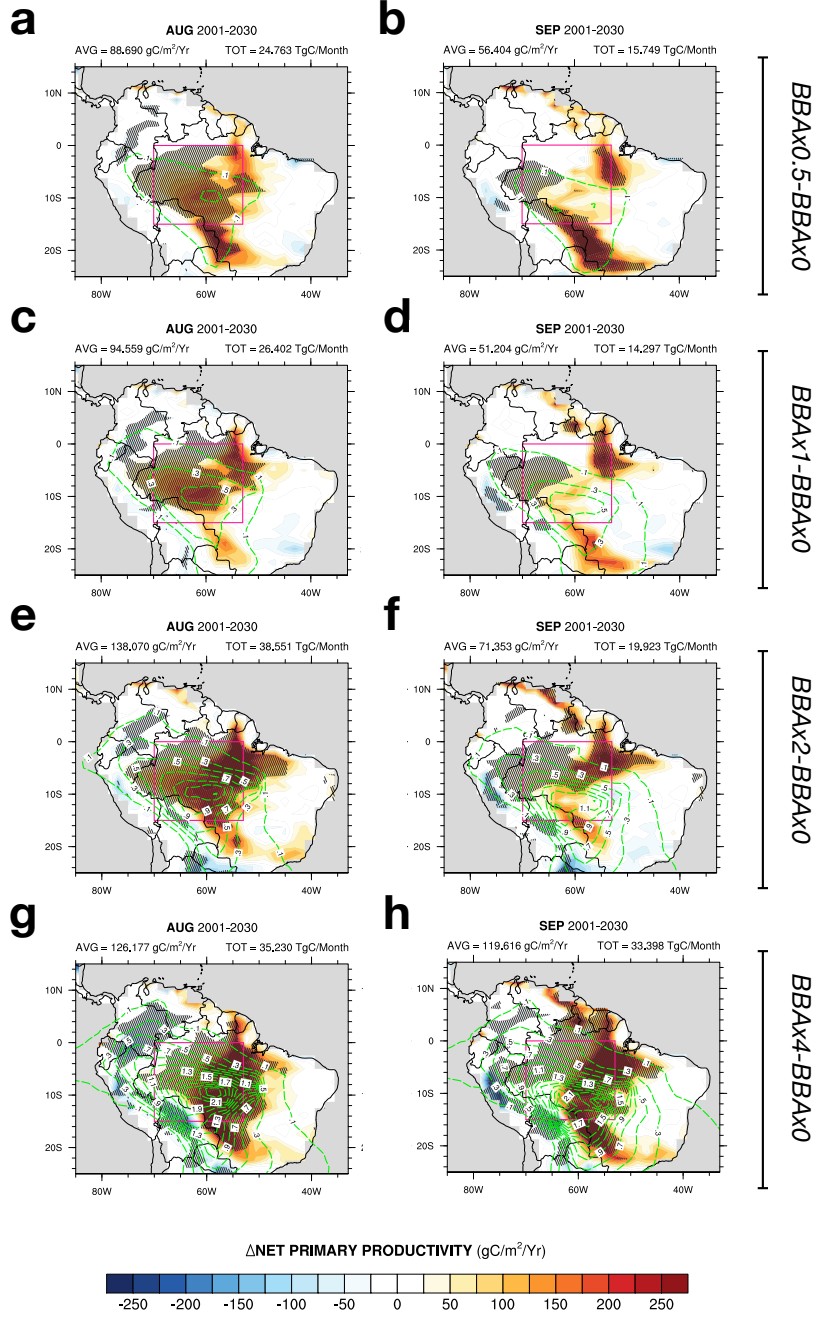

**Figure 5: NPP anomalies (relative to the experiment BBAx0) for the 30 years mean for the four varying BBA emissions (see text, section 2.2) during the August (left) and September (right) months. Mean fluxes (labelled AVG) and accumulation (labelled TOT) are calculated over the domain delimited by the pink borders. Hatched areas represent the regions where changes are significant at the 95% confidence level. Green contours show the 550nm AOT anomalies.**





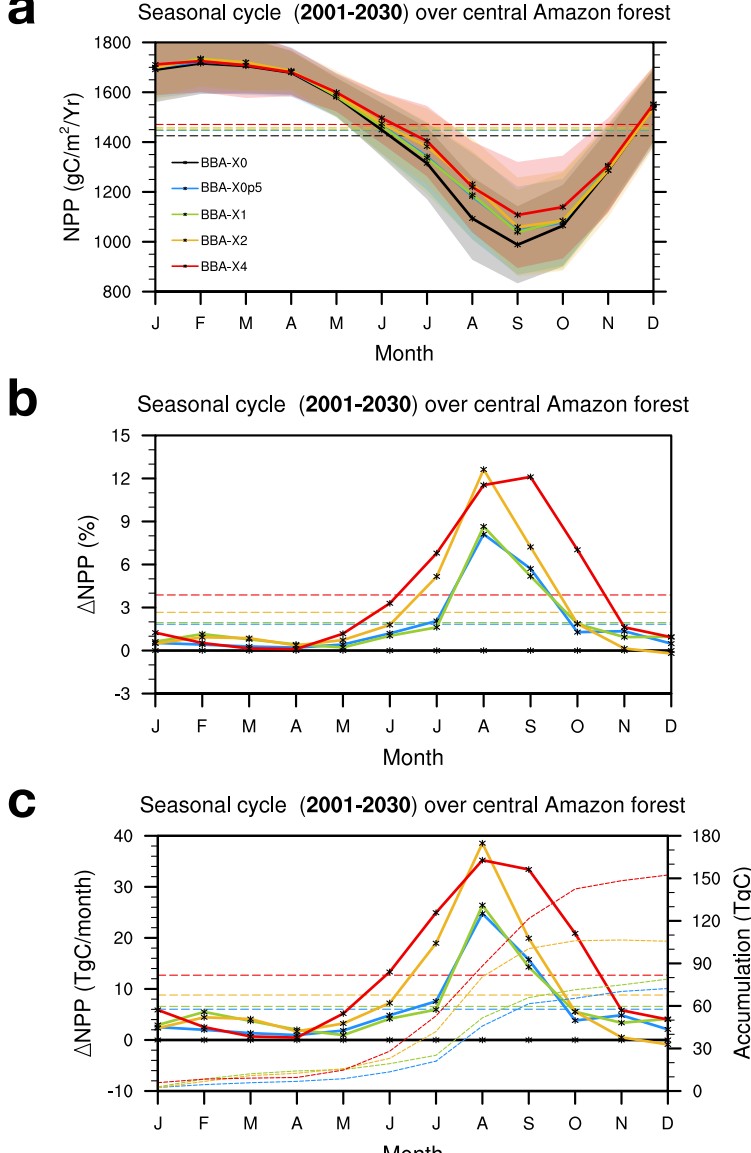

**Figure 6 – Mean seasonal cycle of NPP (a), relative changes (b) and absolute changes (c) for the five BBA emission scenarios (see text, section 2.2) averaged over the Amazon basin. Differences are calculated with regards to experiment BBAx0. The short-dash curves on c) correspond to the accumulated anomalies (right y-axis).**





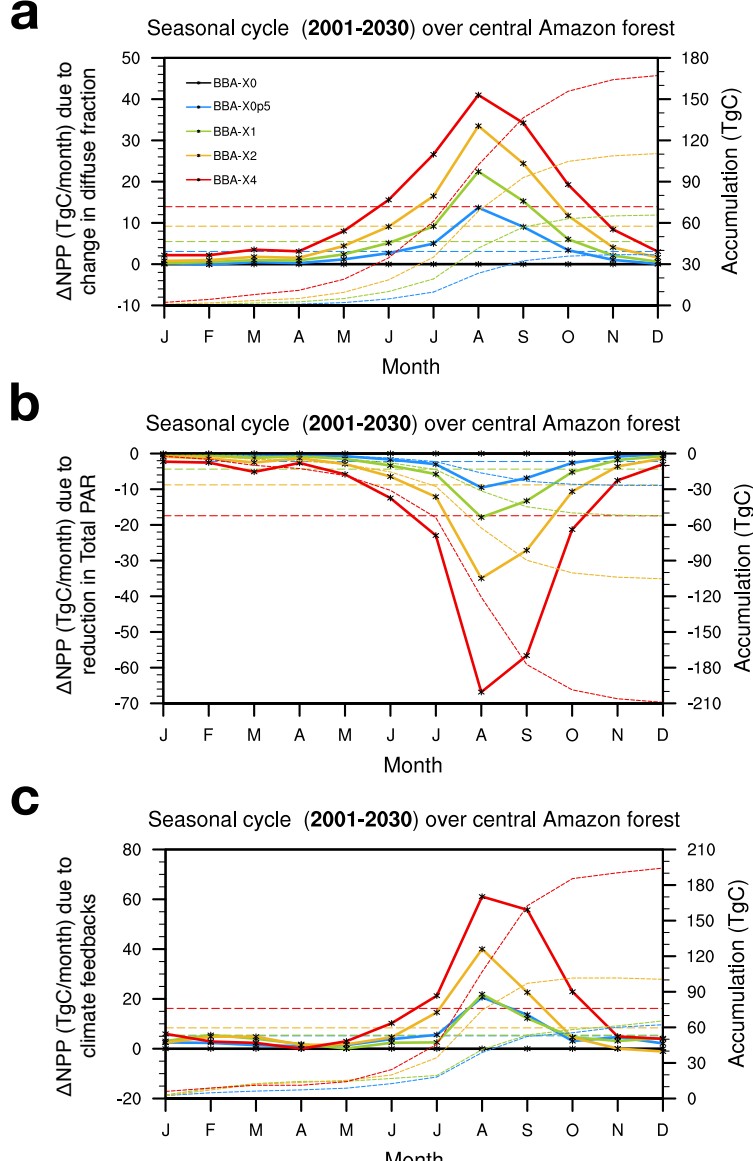

**Figure 7: Similar to Figure 7c, showing the variation in NPP due solely to (a)** *change in diffuse fraction*, **(b)** *reduction in total PAR* **and (c) the** *climate feedback*.



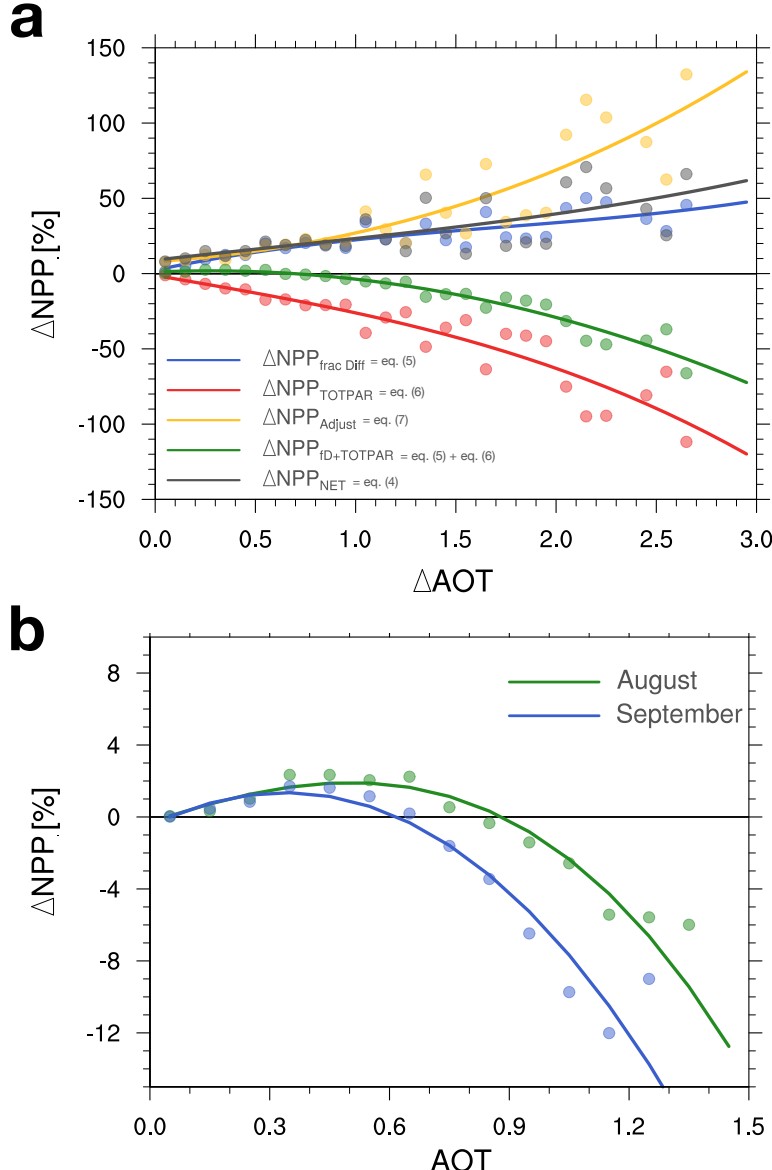

**Figure 8: Showing on a) the relative changes in NPP ($\Delta NPP_{Net}$, in grey), the relative changes in NPP due to the 'change in diffuse fraction' ($\Delta NPP_{Frac\ Diff}$, in blue), 'reduction in total PAR' ($\Delta NPP_{TOTPAR}$, in red), the sum of 'change in diffuse fraction and reduction in total PAR' ($\Delta NPP_{fD+TOTPAR}$, in green, i.e. the DFE) and the climate feedback ($\Delta NPP_{Adjust}$, in yellow) against the anomalies in the AOT at 550nm for the month of August. Showing on b) a zoom on the relative changes in NPP due to 'change in diffuse fraction and reduction in total PAR' ($\Delta NPP_{fD+TOTPAR=DFE}$) – i.e. the changes in NPP only due to change in surface radiation (i.e. the DFE), for August (green) and September (blue) as a function of the total AOT at 550 nm. The dashed lines highlight the AOT thresholds where DFE switch from a positive to a negative impact.**



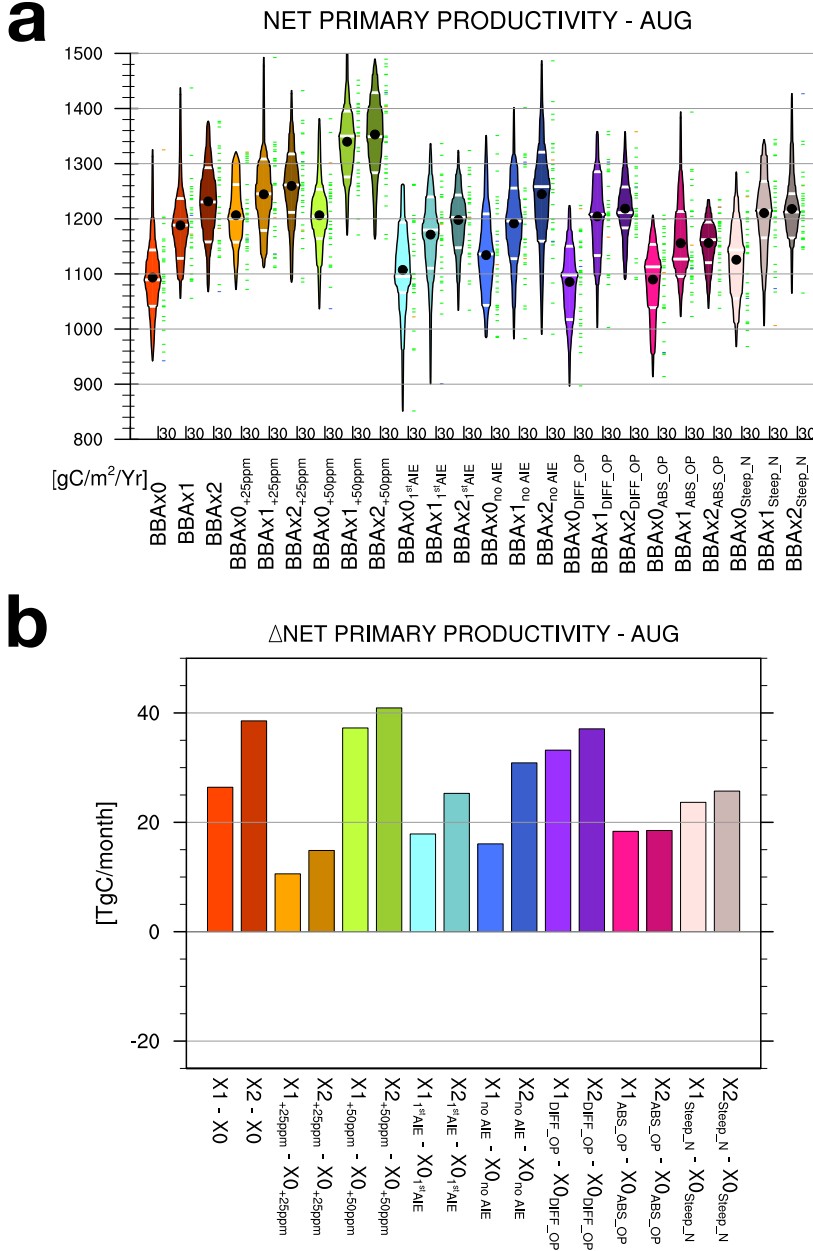

**Figure 9:** Showing on a) a Box and Whiskers plot of the Net Primary Productivity monthly means for August averaged over central Amazon. Result are shown for the main experiment (see text, section 2.2) and the four additional sensitivity experiments (see text, section 2.3). Individual members of the 30 years run are represented by the green dashes. Black dots correspond to the ensemble mean. Dashed white line are the $25^{th}$, $50^{th}$ and $75^{th}$ percentiles. Showing on b) the changes in NPP in each sensitivity experiments, calculated relative to their respective baseline simulation (e.g. $X1_{+25ppm} - X0_{+25ppm}$ is the differences between the BBAx1 and BBAx0 simulations with +25ppm increase in $CO_2$ concentration).