# Peer review of "Studying the impact of biomass burning aerosol radiative and climate effects on the Amazon rainforest productivity with an Earth System Model"

_Atmospheric Chemistry and Physics, 2018_

## Referee Comment (RC1) · Anonymous Referee #1 · 9 Nov 2018

This study uses a coupled modelling framework based on the HadGEM2-ES Earth System model to quantify the effect of Amazon region biomass burning aerosol on the terrestrial carbon cycle through changes in direct and diffuse surface radiation and feedback from climate adjustments. Assessing the ability of Earth System models to fully simulate such effects is very important and this study is a timely and welcome addition to existing work in this area - especially as other coupled model studies investigating diffuse fertilisation effects are based on a different model. The manuscript has a very good structure and is generally well written.

[Figure]

I have a few comments and recommendations that I would like to see addressed before publication.

Specific comments:

1. It is not clear why the authors chose to use the CLASSIC aerosol scheme to represent aerosols in their model, instead of the modal aerosol microphysics scheme (GLOMAP-mode) which has already been implemented, tested and widely used in HadGEM models. As shown in Bellouin et al. (2013), GLOMAP-mode provides a better agreement with aerosol observations and re-analysis products than the CLASSIC scheme. The same study also found substantial differences in aerosol direct radiative forcing estimates between the two schemes, i.e. -0.49 Wm-2 for GLOMAP-mode vs. -0.18 Wm-2 for CLASSIC; such differences are likely to have an important impact on diffuse fertilisation effect estimates. If it is unfeasible to repeat simulations also using the modal scheme, could the authors comment on the potential uncertainty associated with the use of the CLASSIC scheme and maybe even estimate this uncertainty?

2. I might have misread Section 2.2, but according to the first paragraph, year 2000 fire emissions have been used in all simulations. Record low fires have been recorded throughout the Amazon region during year 2000, see e.g. Table 7 in van der Werf et al. (2010): 137 Tg C yr-1 of Southern Hemisphere South American fire emissions, i.e. ~50% of the 271 Tg C yr-1 1997-2009 mean. Please clarify exactly what fire emissions have been used in your simulations (also stating the regional amounts of fire emission). If it is only year 2000, than a detailed discussion is needed on interannual variability and the extent to which the low fires from year 2000 are representative for present day.

3. To what extent is the order of switching off the 3 mechanisms important (Table 3 showing the experimental design)? What is the magnitude of the 3 effects if you estimate them in a different way: for example on page 14, retrieving delta_NPP_clim by contrasting NPPˆBBAx1_clim.aer,Tot.aer,FD.aer with NPPˆBBAx0_clim.aer,Tot.aer,FD.aer?

3a. Regarding your assumption on page 12, line 25, namely "neglecting the interdependency between the three terms", to what extent is this true? Can you confirm with your simulations that the overall effect is the sum of the 3 individual effects?

4. Page 4, lines 12-13 & lines 30-31 and paragraph starting on p 22, l 27: why was the ozone damage effect not included in your simulations? As you mention on page 5 line 2, the Pacifico et al. (2015) study used a similar modelling framework (same model), so one would expect that including the effect should be relatively straight-forward?

Technical corrections:

- p 1, l 23-28: please clearly state the time period (year) the estimated values correspond to.

- p 3, l 21: "did not accounted" -> "did not account".

- p 7, l 11-12: please revise sentence – an "and" is probably missing.

- p8, l 24: what is the reason behind applying the "multiplication factor" only for South American sources?

- p11, eq(1): please explicitly state what you mean by "dL".

- p11, l 26: "analyse" -> "analysis".

- p12, l 24-25: I think deltas are missing, when you want to define "delta f_d", "delta TotPAR" and "delta clim".

- p15, l 12-14: saying that the revised configuration provides a better estimate of global GPP is probably too strong a statement considering the actual values. Also, since your study is restricted to the Amazon region, it would be good to say something about how the simulated GPP/NPP values over this region compare to FLUXCOM, Shao et al (2013) and MODIS. Figure 1 a-d suggests an over-prediction of the model (despite an under-prediction of global GPP)?

- p 15, l 27-30: already described within the figure caption, so no need to also describe within the text what each line represents.

- p19, l 21: a bit confusing, as in the Fig. 8 caption you describe the line as "grey" rather than "black". Again, best to describe figure in the caption and only discuss in in the text.

- p 20, l 17: you might have reversed ABS_OP and DIFF_OP (less/more scattering).

- p20, l 23-27: please give exact values here, as it's not clear what you mean by "significant change" so it's best to have actual values here.

- p20, l 24 and p 21, l 1: "do" -> "does".

- p23, l 18: "quantity" -> "quantify".

- p37, Fig 1 caption: looks like a "(reference)" for the EDMI project is missing.

- p44, Fig 8 caption refers to some missing dashed lines.

References:

Bellouin, N., et al. (2013), Impact of the modal aerosol scheme GLOMAP-mode on aerosol forcing in the Hadley Centre Global Environmental Model, Atmos. Chem. Phys., 13, 3027–3044.

van der Werf, G.R., et al. (2010), Global fire emissions and the contribution of deforestation, savanna, forest, agricultural, and peat fires (1997–2009), Atmos. Chem. Phys., 10, 11707–11735.
* * *

---

## Referee Comment (RC2) · Anonymous Referee #2 · 19 Nov 2018

This study explores the impacts of biomass burning aerosols on forest production in Amazon, taking into account both diffuse fertilization effects and the climatic feedback of fire aerosols. Results show that the benefit of increased diffuse radiation is nearly offset by the reduced total radiation, while climatic effects of fire aerosols make the dominant and positive contributions to the regional carbon uptake. This is an interesting and comprehensive study, providing new perspectives for biosphere-pollution interactions. The authors performed considerable amount of sensitivity experiments to isolate individual factors and to quantify associated uncertainties. Here I have only

some minor comments.

Page 2, Line 23: Not sure Spracklen et al. (2012) provides results of fires. A better reference is Randerson, J. T., et al., The impact of boreal forest fire on climate warming, Science, 314(5802), 1130-1132, 2016.

Page 2, Line 32: "to cite a few" what does this mean?

Page 3, Line 21: "did not accounted" should be "account"

Page 4, Line 1: "only two studies". Not correct. For example, Yue et al. (2017b) also used a fully coupled ESM to quantify aerosol climatic and radiative effects on ecosystem. It's better to say "limited studies".

Page 6, Line 11: "The photosynthesis model is based upon the observed processes", what kind of processes? More details.

Page 6, Line 28: "the tropical French Guyana site". It might be inadequate to calibrate model parameters using data from a single site.

Page 8, Line 3: "30-years" should be "30-year"

Page 17, Line 5: "Fig 5d" should be "Fig 4d", the next line should be "Figs 4a, b, c".

Page 21, Line 28: "fertilisation", it's better to use "fertilization" to be consistent with previous instances.

Figure 2: JAS should be July-August-September

Figure 7: Is that possible to calculate the sensitivity of dNPP to BBA, and to compare the values among different months? These results can tell us the impacts of variant environmental/climatic conditions on fire aerosol-induced NPP perturbations.

---

## Author Comment (AC1) · 19 Dec 2018

**Response to referee comments on: Studying the impact of biomass burning aerosol radiative and climate effects on the Amazon rainforest productivity with an Earth System Model.**

**Ref: acp-2018-924**

Firstly, we would like to thank the reviewers for their time and their constructive comments on the manuscript. We are glad that the reviewers find the paper well-written and of interest to the scientific community. Below please find our responses (in blue) to all reviewer comments (in black). We hope that the updated version of our manuscript is now suitable for publication in Atmospheric, Chemistry and Physics.

All the best,
Florent Malavelle on behalf of all co-authors.

**Anonymous Referee #1**

This study uses a coupled modelling framework based on the HadGEM2-ES Earth System model to quantify the effect of Amazon region biomass burning aerosol on the terrestrial carbon cycle through changes in direct and diffuse surface radiation and feedback from climate adjustments. Assessing the ability of Earth System models to fully simulate such effects is very important and this study is a timely and welcome addition to existing work in this area - especially as other coupled model studies investigating diffuse fertilisation effects are based on a different model. The manuscript has a very good structure and is generally well written.

I have a few comments and recommendations that I would like to see addressed before publication.

**Specific comments:**

**1.** It is not clear why the authors chose to use the CLASSIC aerosol scheme to represent aerosols in their model, instead of the modal aerosol microphysics scheme (GLOMAP-mode) which has already been implemented, tested and widely used in HadGEM models. As shown in Bellouin et al. (2013), GLOMAP-mode provides a better agreement with aerosol observations and re-analysis products than the CLASSIC scheme. The same study also found substantial differences in aerosol direct radiative forcing estimates between the two schemes, i.e. -0.49 Wm-2 for GLOMAP-mode vs. -0.18 Wm-2 for CLASSIC; such differences are likely to have an important impact on diffuse fertilisation effect estimates. If it is unfeasible to repeat simulations also using the modal scheme, could the authors comment on the potential uncertainty associated with the use of the CLASSIC scheme and maybe even estimate this uncertainty?

We used HadGEM2-ES because: *i)* we wanted to be consistent with Pacifico et al. (2015) and *ii)* we wanted a configuration for our Earth System Model that is robust, well characterised and scientifically comparable with past published studies. This wasn't possible with HadGEM3 when our analysis started as HadGEM3 was still in its infancy (one might argue it is still as UKESM will be the main contribution from the UK to CMIP6). HadGEM2-ES still represents the state-of-the-art. GLOMAP-mode was implemented within UKCA in HadGEM3, it is therefore not available in HadGEM2-ES. That is why we used the CLASSIC aerosol scheme in the present study. Back-porting this part of the code is unlikely to be feasible in a timely manner and would be a very time-consuming task and is not considered worthwhile.

Radiative forcing (RF) may not be the ideal metric for discussing pro and cons of GLOMAP-Mode vs CLASSIC here. First these are global averages. Secondly, aerosol RF is RF due to the anthropogenic fraction of the aerosol. Thirdly, RF is calculated as a difference between present-day (PD) and preindustrial (PI). As actually highlighted by Bellouin et al. (2013) "the importance of the 1850 baseline highlights how model skill in predicting present-day aerosol does not guarantee reliable forcing estimates".

That being said, this aspect of choosing a specific aerosol representation (CLASSIC vs GLOMAP-Mode) should not affect our results significantly. As argued in the response to the next comment, to the first order (assuming the representation of vegetation processes is appropriate), investigation of the impact of BBA on vegetation only really requires accurate simulation of AOTs (i.e. the quantity that controls the direct aerosol radiative effect). CLASSIC is totally capable of providing accurate AOTs (as depicted on Fig 2 in the main manuscript) over the region studied and the radiative transfer representation hasn't changed substantially between HadGEM2 and HadGEM3.

An addition point regarding the use of CLASSIC vs GLOMAP mode. Bellouin et al. (2013) used a developmental version of HadGEM3. The use of developmental schemes, while understandable, leads to some different results and conclusions. Indeed, a more recent publication which focusses on biomass burning by Johnson et al. (2016) provides a more up-to-date assessment of the differences between CLASSIC and GLOMAP-mode with a specific focus on biomass burning aerosols. The reviewer should also keep in mind that the differences in radiative forcing that are documented in Bellouin et al (2013) are present-day – pre-industrial; we are focussed on scaling present day emissions, which is rather different. Indeed, Johnson et al (2016) show that the impact of the schemes for biomass burning aerosol in their base form is actually rather little over the Amazonian region. This is evident in Figure 2 and 3 of Johnson et al (2016) which are included below for the reviewer's convenience for annual mean and for September (peak of BB season respectively):

[Figure]

*Fig 2 from Johnson et al. (2016). Annual mean.*

[Figure]

*Fig 3 from Johnson et al. (2016). September mean.*

It is difficult to see significant differences in the AOD, and even more difficult to say which is 'better' when compared against e.g. MODIS observations over South America (Fig 3 from Johnson et al. 2016).

[Figure]

*Fig 3 from Johnson et al. (2016). Comparison of CLASSIC and GLOMAP-mode total aerosol compared against C5 and C6 data from MODIS (Johnson et al., 2016)*

Johnson et al. (2016) also evaluate difference in e.g., the single scattering albedo etc between CLASSIC and GLOMAP-Mode. Owing to updates in absorption properties and formulation in the schemes, differences are relatively marginal. Generally CLASSIC is able to represent aerosol direct effects with fidelity because, although it is a single moment scheme (prognostic mass only), aerosol microphysical properties and hence the optical parameters are based on aircraft-based observations of biomass burning aerosol. However, we agree with the reviewer that the indirect radiative effects from single moment schemes can be radically different from those from dual moment schemes like GLOMAP-mode (as increases in aerosol mass do not necessarily increase the aerosol number and hence CCN, Malavelle et al. 2017).

- *Bellouin, N., et al., Impact of the modal aerosol scheme GLOMAP-mode on aerosol forcing in the Hadley Centre Global Environmental Model, Atmos. Chem. Phys., 13, 3027–3044, (2013).*
- *Johnson, B. T., Haywood, J. M., Langridge, J. M., Darbyshire, E., Morgan, W. T., Szpek, K., Brooke, J. K., Marenco, F., Coe, H., Artaxo, P., Longo, K. M., Mulcahy, J. P., Mann, G. W., Dalvi, M., and Bellouin, N.: Evaluation of biomass burning aerosols in the HadGEM3 climate model with observations from the SAMBBA field campaign, Atmos. Chem. Phys., 16, 14657-14685, https://doi.org/10.5194/acp-16-14657-2016, (2016).*
- *Malavelle, F. F. et al., Strong constraints on aerosol–cloud interactions from volcanic eruptions, Nature, 546, (2017).*

**2.** I might have misread Section 2.2, but according to the first paragraph, year 2000 fire emissions have been used in all simulations. Record low fires have been recorded throughout the Amazon region during year 2000, see e.g. Table 7 in van der Werf et al. (2010): 137 Tg C yr-1 of Southern Hemisphere South American fire emissions, i.e. ~50% of the 271 Tg C yr-1 1997-2009 mean. Please clarify exactly what fire emissions have been used in your simulations (also stating the regional amounts of fire emission). If it is only year 2000, than a detailed discussion is needed on interannual variability and the extent to which the low fires from year 2000 are representative for present day.

The BBA emissions used here are not the emissions for the exact year 2000 but a decadal mean centred on 2000. This is mentioned at P8, Lines15-16:

*"Aerosol and their precursor emissions are taken from the CMIP5 inventories (Lamarque et al., 2010). We use the decadal mean emissions centred around the year 2000 representative of present-day emissions"*

The biomass burning emissions used during CMIP5 are based on the GFEDv2 inventory (van der Werf et al., 2006) for the 1997–2006 period. As detailed in Lamarque et al. (2010): *"Given the substantial interannual variability of biomass burning on a global and regional scale (e.g., Duncan et al., 2003; Schultz et al., 2008), it is problematic to use a snapshot dataset from an individual year for the development of a dataset that is considered to be representative for a decade. [...] for the 2000 estimate which is calculated from the 1997–2006 average."*

We have modified P8, Lines20-23 to make this clearer:

*"Aerosol and their precursor emissions are the dataset used during CMIP5 (Lamarque et al., 2010). [ ... ] Given the substantial interannual variability of biomass burning on a global and regional scale, a present-day climatology (i.e. average year) is calculated as the GFEDv2 1997-2006 average (Lamarque et al., 2010).*

Note that the reason for varying aerosol emissions was to account for the variability in BBA sources and intensity. These variations are readily visible from the MODIS AOT retrievals when looking at the individual years between 2001 and 2016 (Fig. 2b in the main manuscript). The emission multiplication factors were therefore considered such as HadGEM2 could represent a similar range of AOTs. AOT is the quantity used by the radiative transfer and is responsible of the aerosol direct radiative effects which eventually affect vegetation. We therefore favoured constraining BBA via the 'optics' in HadGEM2 rather than via emissions as discrepancies in modelled particulate matter and modelled AOT are a common feature of aerosol models (including GLOMAP-Mode, e.g. Reddington *et al.* 2016, 2018).

- *Reddington, C. L., Spracklen, D. V., Artaxo, P., Ridley, D. A., Rizzo, L. V., and Arana, A.: Analysis of particulate emissions from tropical biomass burning using a global aerosol model and long-term surface observations, Atmos. Chem. Phys., 16, 11083-11106, https://doi.org/10.5194/acp-16-11083-2016, 2016.*
- *Reddington, C. L., Morgan, W. T., Darbyshire, E., Brito, J., Coe, H., Artaxo, P., Marsham, J., and Spracklen, D. V.: Biomass burning aerosol over the Amazon: analysis of aircraft, surface and satellite observations using a global aerosol model, Atmos. Chem. Phys. Discuss., https://doi.org/10.5194/acp-2018-849, in review, 2018.*

**3a.** To what extent is the order of switching off the 3 mechanisms important (Table 3 showing the experimental design)? What is the magnitude of the 3 effects if you estimate them in a different way: for example on page 14, retrieving delta_NPP_clim by contrasting NPP^BBAx1_clim.aer,Tot.aer,FD.aer with NPP^BBAx0_clim.aer,Tot.aer,FD.aer?

This is a good question. It will be addressed in the next comment as our answer covers both comments 3a and 3b.

**3b.** Regarding your assumption on page 12, line 25, namely "neglecting the interdependency between the three terms", to what extent is this true? Can you confirm with your simulations that the overall effect is the sum of the 3 individual effects?

This is a very good question too. We answer this as follows:

We will start by answering the suggestion of using the term **[NPP^BBAx1_clim.aer,TotPAR.aer,F$_d$.aer - NPP^BBAx0_clim.aer,TotPAR.aer,F$_d$.aer]** as an alternative way to retrieve d_NPP_clim (comment 3a). This term actually corresponds to the net effect of BBA on the vegetation (i.e. first term on the LHS in eq. 4), so it includes both the effect of the fast climate adjustments and effect of changes in radiation due to the BBA.

The equation 3 from the main manuscript is reproduced below:

$$\delta NPP \cong \frac{\partial NPP}{\partial f_d}\delta f_d + \frac{\partial NPP}{\partial TotPAR}\delta TotPAR + \frac{\partial NPP}{\partial Clim}\delta Clim$$

Note that we used the symbol $\cong$ instead of a strict equality as we have neglected the higher order terms here. This is now written this way in the manuscript.

This equation can be interpreted in an Effective Radiative Forcing (ERF) framework. The net change in NPP (the term on LHS) is the result of the contribution of *i)* the aerosol direct radiative effect (i.e. the change in radiation seen by the vegetation represented by the two first terms on the RHS) and *ii)* the fast climate adjustments due to the aerosol forcing (i.e. the 3$^{rd}$ term on the RHS). In a condensed form, eq. 3 can be rewritten as follow:

$$\delta NPP \cong \frac{\partial NPP}{\partial \text{Radiation}} \delta Radiation + \frac{\partial NPP}{\partial Clim} \delta Clim$$

Where Radiation contains the effect of changes in TotPAR and the diffuse fraction ($f_d$). The two contributions ($\delta Radiation$ and $\delta Clim$) can be assessed independently although the changes of the climate can slightly affect the values of $f_d$ and TotPAR (e.g. via changes in cloudiness, more on this later).

The first term ($\delta Radiation$) is calculated at each model time-step (i.e. for a fixed climate) whereas the second term ($\delta Clim$) is calculated from a pair of simulations where the effect of aerosols on the radiation seen by the vegetation are not considered (i.e. $TotPAR = clean, Fd = clean$).

The contribution of the fast climate adjustments to the aerosol forcing ($\delta Clim$) is evaluated using eq. 7. An alternative writing for eq. 7 in numerical form that is consistent with eq. 5 and eq. 6 is as follows:

$$\Delta \overline{NPP}_{clim}^{BBAx1} = \left\{ \overline{NPP}_{Clim.aer,Tot.clean,Fd.clean}^{BBAx1} - \overline{NPP}_{Clim.clean,Tot.clean,Fd.clean}^{BBAx1} \right\} \\ - \left\{ \overline{NPP}_{Clim.aer,Tot.clean,Fd.clean}^{BBAx0} - \overline{NPP}_{Clim.clean,Tot.clean,Fd.clean}^{BBAx0} \right\}$$

As clim.clean corresponds to the climate that has not experienced any aerosol radiative forcing, the terms $\overline{NPP}_{Clim.clean,Tot.clean,Fd.clean}^{BBAx1}$ and $\overline{NPP}_{Clim.clean,Tot.clean,Fd.clean}^{BBAx0}$ are effectively equal. Therefore the climate effect of BBA on vegetation can be simplified to retrieve eq. 7:

$$\Delta \overline{NPP}_{clim}^{BBAx1} = \left( \overline{NPP}_{Clim.aer,Tot.clean,Fd.clean}^{BBAx1} \right) - \left( \overline{NPP}_{Clim.aer,Tot.clean,Fd.clean}^{BBAx0} \right)$$

We believe this is the cleanest estimate of the impact of climate adjustments due to the BBA forcing on the vegetation productivity.

Now remains the calculation of the contribution from $\delta Radiation$. In the manuscript we assumed a first order expansion so we can evaluate independently the contributions from $f_d$ and TotPAR, namely:

$$\frac{\partial NPP}{\partial Radiation} \delta Radiation \cong \frac{\partial NPP}{\partial f_d} \delta f_d + \frac{\partial NPP}{\partial TotPAR} \delta TotPAR = F + T$$

Where F represents the effect of $f_d$ and T the effect of TotPAR. In order to get the most accurate estimation of F and T, the order of switching on/off the 2 mechanisms is indeed important. We calculate the effect of the reduction in TotPAR first. Then we calculate the effect of the increase in $f_d$. Doing the other way around would give too much weight to the DFE as a high $f_d$ with a high TotPAR would increase the vegetation productivity unrealistically (i.e. an increase in $f_d$ is always concomitant with a decrease in TotPAR).

In equation 3 (see beginning of this lengthy response), we neglected the dependency between $f_d$ and TotPAR and Clim. The exact value of the contribution of $\delta Radiation$ could therefore be written as:

$$\frac{\partial NPP}{\partial Radiation} \delta Radiation = \frac{\partial NPP}{\partial f_d} \delta f_d + \frac{\partial NPP}{\partial TotPAR} \delta TotPAR + R = TRUE\_dRAD$$

Where R corresponds to the higher order crossed terms between $f_d$ and TotPAR and Clim.

In the manuscript, we provided estimates of F and T. If this first order linearization is a suitable approximation, then the sum of F and T should be close to TRUE_dRAD (i.e. R is small).

With our diagnostics we are able to calculate both TRUE_dRAD (i.e. by summing the effect of $f_d$ and TotPAR before temporal and spatial averaging) and the sum of F + T (i.e. summing the temporally and spatially averaged F and T). These calculations have been done for the NPP over the domain of analysis defined in the manuscript. The results are for the month of August but these hold for the other months. Results are shown in the table below:

| d_NPP (wrt BBAX0) in TgC/month | F | T | F+T | TRUE_dRAD | R |
|---|---|---|---|---|---|
| BBAX0.5 | 13.7 | -9.6 | 4.1 | 5.0 | 0.9 |
| BBAX1 | 22.4 | -17.9 | 4.5 | 5.1 | 0.6 |
| BBAX2 | 33.5 | -35.0 | -1.5 | -2.5 | -1.0 |
| BBAX4 | 41.0 | -66.8 | -25.8 | -34.6 | -8.8 |

The agreement between F+T and TRUE_dRAD is reasonable which supports that a first order linearisation is an appropriate approximation. It becomes criticisable for the high aerosol simulation (BBAX4) however. Mathematically, it means that the linearisation starts to be inaccurate as the deviation from the reference point becomes large. Physically it can be explained by these two mechanisms:

- As an increase in $f_d$ is always concomitant with a decrease in TotPAR, the two variables are not truly independent.
- If the change in climate is important, there is the possibility that it affects $f_d$ and ToTPAR (e.g. through change in cloudiness via aerosol semi-direct effects).

In summary, when AOTs get very high, the error between F+T and TRUE_dRAD is likely to increase.

If the two contributions from $\delta Radiation$ and $\delta Clim$ were truly independent, then adding them together should give us the same amount of change in NPP as when calculating the changes from eq. 4. To verify this, we calculated the budgets in the similar way as the table above:

| d_NPP (wrt BBAX0) in TgC/month | TRUE_dRAD | $\delta Clim$ (eq. 7) | TRUE_dRAD + $\delta Clim$ | NET d_NPP (eq. 4) | Differences (i.e. ~R) |
|---|---|---|---|---|---|
| BBAX0.5 | 5.0 | 20.6 | 25.6 | 24.8 | -0.8 |
| BBAX1 | 5.1 | 21.9 | 27.0 | 26.4 | -0.6 |
| BBAX2 | -2.5 | 40.0 | 37.5 | 38.6 | 0.9 |
| BBAX4 | -34.6 | 61.1 | 26.5 | 35.3 | 8.8 |

Understanding the effect of $\delta clim$ on $\delta Radiation$ (e.g. via change in cloudiness) would certainly be an interesting academic problem but it would be way beyond the scope of this paper and not feasible with the set of simulations we have conducted. Besides this seems to be of second order (as $\delta Clim >>$ TRUE_dRAD unless AOT is large).

The bottom line is that our conclusions remain unchanged; we can separate the contribution of the three terms safely. Overall, the variation in radiation are contributing much less to the changes in NPP than the climate adjustments resulting from the effect of BBA.

**4.** Page 4, lines 12-13 & lines 30-31 and paragraph starting on p 22, l 27: why was the ozone damage effect not included in your simulations? As you mention on page 5 line 2, the Pacifico et al. (2015) study used a similar modelling framework (same model), so one would expect that including the effect should be relatively straight-forward?

Indeed, it would be easy to add the ozone effect in the current study. The two studies (Pacifico et al, and this one) were conducted in parallel for the same project therefore these were addressing two scientific questions separately in order to construct a good understanding of the respective mechanisms. Besides, ozone and aerosols effects will have a different spatial footprint (due to the chemistry of Ozone), which warrants studying them independently before. We are now analysing new ESM simulations at the global scale which combine both the DFE and the Ozone damage. We plan to submit these results for publication in a separate manuscript.

**Technical corrections:**

- p 1, l 23-28: please clearly state the time period (year) the estimated values correspond to.

Added: "*Results show that the overall net impact of present-day, defined as year 2000 climate, biomass burning aerosols is to increase net primary productivity (NPP) by …*"

Note that as explained in the method section, we use climatological forcing (including the BBA emissions) so we do not specifically simulate a given year but a period representative of the mean state of the Earth climate.

- p 3, l 21: "did not accounted" -> "did not account".

Modified.

- p 7, l 11-12: please revise sentence – an "and" is probably missing.

Indeed, sentence altered: "*Transported species experience boundary layer and convective mixing, and are removed by dry and wet deposition*"

- p8, l 24: what is the reason behind applying the "multiplication factor" only for South American sources?

Two reasons:

i)      We wanted to be consistent with the Pacifico et al., 2015 as explained above.
ii)     We also multiplied BBA emissions globally in separate simulations. In doing so, we noticed that the contribution from Southern African fires over the Amazon was non-negligible. In addition, a strong increase in global BBA emissions (e.g. X2 and X4) introduces additional effects on the global climate such as expansion of the Hadley cells (due to the absorbing nature of these aerosols).

- p11, eq(1): please explicitly state what you mean by "dL". - p11, l 26: "analyse" -> "analysis".

dL was removed from the equation, that was inconsistent writing of the exponential decay profile. Notation for the leaf level Nitrogen was slightly altered for improved clarity:

$$N_{Leaf}(\mathrm{L}) = N_{L0}e^{-K_N L}$$

- p12, l 24-25: I think deltas are missing, when you want to define "delta f_d", "delta TotPAR" and "delta clim".

In the way the sentence was originally written, you are right and the deltas should appear. However, the aim was to define the nomenclature that is used in the remainder of the manuscript (e.g. $f_d$ for diffuse fraction). As such we slightly modified the sentence:

"*A simple theoretical framework can be used to discriminate a fast carbon flux, e.g. NPP, as a function of the 'diffuse fraction', $f_d$, the 'total PAR', TotPAR and the 'climate feedback', clim, such as NPP($f_d$, TotPAR, clim).*"

- p15, l 12-14: saying that the revised configuration provides a better estimate of global GPP is probably too strong a statement considering the actual values. Also, since your study is restricted to the Amazon region, it would be good to say something about how the simulated GPP/NPP values over this region compare to FLUXCOM, Shao et al (2013) and MODIS. Figure 1 a-d suggests an over-prediction of the model (despite an under-prediction of global GPP)?

This is a fair point. If we were to assume that the best estimate sits somewhere in between FLUXCOM (+129 PgC/yr) and for Shao et al. (2013, +118 PgC/yr), then the updated GPP estimate in HadGEM2 (i.e.

+115 PgC/yr) sits slightly closer than the estimate from the original model configuration (+140 PgC/yr). Indeed, we can argue that the underestimation of GPP in the updated HadGEM2 configuration is comparable (in magnitude) to the overestimation of GPP in the original HadGEM2 configuration. We hinted towards 'better' because the ratio of NPP over GPP in the updated version of HadGEM2 is more consistent with observationally-based ratio estimates (e.g. Luyssaert et al., 2007) whereas the original HadGEM2 configuration had too low NPP/GPP ratios. Given the large uncertainties in all these estimates, 'better' is probably a bit of a stretch so we have modified the text accordingly. In addition we quantified the GPP more specifically for the Amazon region as suggested.

*"The underestimation of the GPP in the updated HadGEM2-ES configuration is comparable in magnitude to the overestimation of the GPP in the HadGEM2-ES configuration. However, the ratio of NPP over GPP (not shown) in the updated version of HadGEM2 is more consistent with observationally-based ratio estimates (e.g. Luyssaert et al., 2007). Despite the inherent uncertainties in the two reference estimates of the global GPP (i.e. between +118 and 129 TgC/yr), it suggests that the updated version of HadGEM2-ES is able to provide a more consistent global GPP estimate. Over the central Amazon domain which is represented by the region encapsulated in the red box on Fig. 2a., HadGEM2-ES averaged GPP in August (respectively September) is 2750 ± 250 gC/m²/yr (respectively 2600 ± 200 gC/m²/yr for September) compared to 2250 ± 125 gC/m²/yr (respectively 2500 ± 180 gC/m²/s for September) for FLUXCOM. [...] Despite obvious overestimation by HadGEM2-ES of the NPP on annual mean over South America when compared to MOD17A2 (Fig. 1b and 1d) the fluxes are well captured during the peak of the fire season over the central Amazon. The average GPP from HadGEM2-ES in August (respectively September) is 1080 ± 140 gC/m²/yr (respectively 975 ± 100 gC/m²/yr for September) compared to 990 ± 550 gC/m²/yr (respectively 1025 ± 590 gC/m²/s for September) for MOD17A2."*

- p 15, l 27-30: already described within the figure caption, so no need to also describe within the text what each line represents.

Redundant figure description removed from main text.

- p19, l 21: a bit confusing, as in the Fig. 8 caption you describe the line as "grey" rather than "black". Again, best to describe figure in the caption and only discuss in in the text.

Redundant figure description removed from main text. We changed the colour name to 'dark grey' in the legend of fig. 8.

- p 20, l 17: you might have reversed ABS_OP and DIFF_OP (less/more scattering).

Well spotted. It has been corrected.

- p20, l 23-27: please give exact values here, as it's not clear what you mean by "significant change" so it's best to have actual values here.

We have added a discussion with more quantified substance:

*"However, we do not observe a significant change in the modelled BBA impact on vegetation productivity for the varying BBA scattering/absorbing assumptions (Fig. 9b). In the standard simulations, the net change in NPP due to BBA is +28.4 to 38.6 TgC/month in August. For the DIFF_OP simulation (respectively ABS_OP) the net change in NPP is +32.1 to 36.2 TgC/month (respectively +17.9 to 18.2 TgC/month). For September (not shown), we actually found that the ABS_OP simulation had the largest increase in NPP which is not consistent with our assumption. In summary, the effect of BBA optical properties on NPP changes are within the noise and considered negligible. This can be explained in the light of the results discussed in Sect. 3.3, where we showed that the DFE from present-day BBA is small (~ +5 TgC/month in August in BBAX1) for this model in this region of the world. Therefore, altering the ratio of diffuse fraction reaching the ground via the aerosol optical properties, that is modulating the magnitude of the DFE, does not have a measurable effect on vegetation productivity."*

-p20, l24 and p21, l1: "do"->"does".

corrected

- p23, l 18: "quantity" -> "quantify".

corrected

- p37, Fig 1 caption: looks like a "(reference)" for the EDMI project is missing.

I've removed that '(reference)' from the legend. EMDI data are accessible online. The link to the dataset is provided in the main manuscript (P14, line 13) and in the data availability section.

- p44, Fig 8 caption refers to some missing dashed lines.

Figure corrected. Also changed the axis labels on Fig. 8 for improved clarity. Note that figure 8b is plotted against the total AOT. This is now mentioned more clearly in the text P20, lines 0-5.

References:

Bellouin, N., et al. (2013), Impact of the modal aerosol scheme GLOMAP-mode on aerosol forcing in the Hadley Centre Global Environmental Model, Atmos. Chem. Phys., 13, 3027–3044.

van der Werf, G.R., et al. (2010), Global fire emissions and the contribution of de- forestation, savanna, forest, agricultural, and peat fires (1997–2009), Atmos. Chem. Phys., 10, 11707–11735.

**Anonymous Referee #2**

This study explores the impacts of biomass burning aerosols on forest production in Amazon, taking into account both diffuse fertilization effects and the climatic feedback of fire aerosols. Results show that the benefit of increased diffuse radiation is nearly offset by the reduced total radiation, while climatic effects of fire aerosols make the dominant and positive contributions to the regional carbon uptake. This is an interesting and comprehensive study, providing new perspectives for biosphere-pollution interactions. The authors performed considerable amount of sensitivity experiments to isolate individual factors and to quantify associated uncertainties. Here I have only some minor comments.

Page 2, Line 23: Not sure Spracklen et al. (2012) provides results of fires. A better reference is Randerson, J. T., et al., The impact of boreal forest fire on climate warming, Science, 314(5802), 1130-1132, 2016.

Good point, Spracken et al. 2012 was about analysing the water content of air masses that experienced vegetated area along transport. We were thinking of more recent work from D. Spracken's group however this is not the most suitable reference to support the point made in this sentence. We used Zemp et al. (2017) instead of Randerson et al (2016) as the focus is more on tropical forests.

- *Zemp, D. C. et al.: Self-amplified Amazon forest loss due to vegetation–atmosphere feedbacks. Nat. Commun. 8, 14681 doi:10.1038/ncomms14681 (2017).*

Page 2, Line 32: "to cite a few" what does this mean?

Rephrased:

"*Assessing the overall impact of Amazonian forest fires on ecosystems is challenging as it encompasses a combination of direct losses, and indirect impacts from the fire by-products which can depend on intricate interactions among several earth system components, including: the biosphere, atmospheric composition, radiation and energy budget, clouds and the water cycle (Bonan 2008).*"

Page 3, Line 21: "did not accounted" should be "account"

Corrected.

Page 4, Line 1: "only two studies". Not correct. For example, Yue et al. (2017b) also used a fully coupled ESM to quantify aerosol climatic and radiative effects on ecosystem. It's better to say "limited studies".

Agree that's better wording. As a consequence, the next sentence has been modified to take into account Yuan et al. (2007b) findings:

"*Only a limited number of studies have considered the DFE within a fully coupled earth system framework (e.g. Strada and Unger, 2016; Unger et al., 2017, Yue et al., 2017b using the NASA GISS ModelE2–YIBs) to investigate the role of aerosols and haze on vegetation. Although these studies have investigated the role of diffuse radiation on GPP and isoprene emissions (Strada and Unger, 2016; Unger et al., 2017), understanding of the indirect impact of climate effects from aerosols on vegetation productivity remains very uncertain. This was addressed over China by Yue et al. (2007b) who demonstrated that aerosol induced hydroclimatic feedbacks can promote ecosystem NPP.*"

Page 6, Line 11: "The photosynthesis model is based upon the observed processes", what kind of processes? More details.

Added the processes in the sentence. Note that full description of these are provided by Collatz et al. (1991, 1992) which is referenced at the end of this paragraph:

"The photosynthesis model is based upon the observed processes of gas and energy exchange at the leaf scale, which are then scaled up to represent the canopy. It takes into account variations in direct and diffuse radiation on sunlit and shaded canopy photosynthesis at each canopy layer. In this way, photosynthesis of sunlit and shaded leaves is calculated separately under the assumption that shaded leaves receive only diffuse light and sunlit leaves receive both diffuse and direct radiation (Dai et al., 2004; Clark et al., 2011). Leaf-level photosynthesis is calculated using the biochemistry of C3 and C4 photosynthesis from Collatz et al. (1991) and Collatz et al. (1992)."

Page 6, Line 28: "the tropical French Guyana site". It might be inadequate to calibrate model parameters using data from a single site.

We totally agree. Unfortunately, good quality observations that include measurement of total and diffuse light in addition to carbon fluxes in the Amazon are rare. Note that this exercise (modelling of the French Guyana site) is not for validation purposes but for exploring JULES sensitivity to parameters using realistic forcing. Evaluation of GPP/NPP is provided by comparing HadGEM2 against MODIS and FLUXCOM in the result section.

Page 8, Line 3: "30-years" should be "30-year"

Modified 30-years to 30 years to be consistent with other part of the text.

Page 17, Line 5: "Fig 5d" should be "Fig 4d", the next line should be "Figs 4a, b, c".

Well spotted. Also corrected on Page 16, Line 32.

Page 21, Line 28: "fertilisation", it's better to use "fertilization" to be consistent with previous instances.

Good point about consistency. ACP accepts all standard varieties of English in order to retain the author's voice. However, the variety should be consistent within each article. Because we have used British English in the rest of the manuscript, we have replaced occurrences of Fertilization with Fertilisation.

Figure 2: JAS should be July-August-September

Corrected.

Figure 7: Is that possible to calculate the sensitivity of dNPP to BBA, and to compare the values among different months? These results can tell us the impacts of variant environmental/climatic conditions on fire aerosol-induced NPP perturbations.

I'm not sure that we understand what is suggested. Do you suggest calculating NPP susceptibility to BBA (e.g. d ln(NPP) / d ln (BBA_emissions) ) ?

Note that the vast majority of BBA emissions over the Amazon occur during the dry season (peaking in August and September) so it is very likely that we would not be able to derive a clear signal outside that period.